# ADAPTIVE VISUAL SCENE UNDERSTANDING: INCREMENTAL SCENE GRAPH GENERATION

## ABSTRACT

Scene graph generation (SGG) analyzes images to extract meaningful information about objects and their relationships. In the dynamic visual world, it is crucial for AI systems to continuously detect new objects and establish their relationships with existing ones. Recently, numerous studies have focused on continual learning within the domains of object detection and image recognition. However, a limited amount of research focuses on continual learning specifically in the context of SGG. This increased complexity arises from the intricate interactions among objects, their associated contexts, and the dynamic relationships among these objects. Investigating continual learning in scene graph generation becomes particularly challenging due to the necessity to expand, modify, retain, and reason scene graphs within the process of adaptive visual scene understanding. To systematically explore Continual Scene Graph Generation (CSEGG), we present a comprehensive benchmark comprising three learning regimes: relationship incremental, scene incremental, and relationship generalization regimes. Moreover, we introduce a "Replays via Analysis by Synthesis" strategy named RAS. This approach leverages the scene graphs, decomposes and re-composes them to represent different scenes, and replays the synthesized scenes based on these compositional scene graphs. The replayed synthesized scenes act as a means to practice and refine proficiency in SGG in known and unknown environments. Our experimental results not only highlight the challenges of directly combining existing continual learning methods with SGG backbones but also demonstrate the effectiveness of our proposed approach, enhancing CSEGG efficiency while simultaneously preserving privacy. We will release our code and data upon publication.

## 1 INTRODUCTION

Scene graph generation (SGG) aims to extract object entities and their relationships in a scene. The resulting scene graph, carrying semantic scene structures, can be used for a variety of downstream tasks such as object detection(Szegedy et al., 2013), image captioning (Hassan et al., 2023; Aditya et al., 2015) , and visual question answering (Ghosh et al., 2019). Despite the notable advancements in SGG, current works have largely overlooked the critical aspect of continual learning. In the dynamic visual world, new objects and relationships are introduced incrementally, posing challenges for SGG models to account for new changes without forgetting previously acquired knowledge. This problem of Continual ScenE Graph Generation (CSEGG) holds great potential for various applications, such as real-time robotic navigation in dynamic environments and adaptive augmented reality experiences.

The field of continual learning has witnessed significant growth in recent years, with a major focus on tasks such as image classification (Mai et al., 2021), object detection (Wang et al., 2021a), and visual question answering (Lei et al., 2022). However, these endeavors have largely neglected the distinctive complexities associated with CSEGG. Here, we highlight several unique challenges of CSEGG: (1) In contrast to object detection, SGG involves understanding and capturing the relationships between objects, which can be intricate and diverse. Consequently, in CSEGG, conveying the spatial and semantic relationships between objects demands adaptive reasoning from the dynamic scene. (2) SGG introduces a higher level of combinatorial complexity than object detection and image classification because each detected object pair may have multiple potential spatial and functional relationships. Thus, as new objects are introduced to the scenes, the complexity of relationships among all the objects increases significantly in a non-linear fashion. (3) The long-tailed distribution in SGG can be attributed to the inherent characteristics of real-world scenes, where certain objects are more prevalent than others. Consequently, CSEGG requires the computational models to adapt continually to the evolving long-tailed distributions over different scenes. Due to a scarcity of research specifically

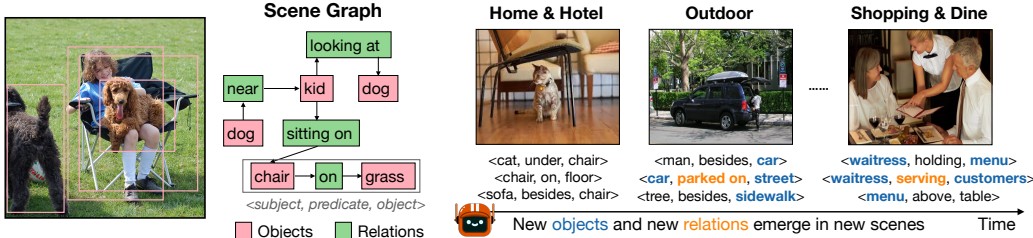

(a) Scene Graph Generation (SGG)  (b) Continual ScenE Graph Generation (CSEGG)

Figure 1: (a) **A scene graph is a graph structure,** where objects are represented as nodes (red boxes), and the relationships between objects are represented as edges connecting the corresponding nodes (green boxes). Each node in the graph contains information such as the object's class label, and spatial location. The edges in the graph indicate the relationships between objects, often described by predicates. A scene graph can be parsed into a set of triplets, consisting of three components: a subject, a relationship predicate, and an object that serves as the target or object of the relationship. The graph allows for a compact and structured representation of the objects and their relationships within a visual scene. (b) **An example CSEGG application** is presented, where a robot continuously encounters new objects (blue) and new relationships (yellow) over time across new scenes.

addressing these challenges of CSEGG, there is a pressing need for specialized investigations and methodologies to enable computational models with the ability of CSEGG.

In this study, we re-organize existing SGG datasets (Krishna et al., 2017; Kuznetsova et al., 2020) to establish a novel and comprehensive CSEGG benchmark with 3 learning protocols as shown in **Fig. S2**. **(S1).** Relationship-incremental setting: an SGG agent learns to recognize new relationships among familiar objects within the same scene. **(S2).** Scene-incremental setting: an SGG agent is deployed in new scenes where it has to jointly learn to detect new objects and classify new relationships. **(S3).** Relationship generalization setting: an SGG agent generalizes to recognize known relationships among unknown objects, as the agent learns to recognize new objects.

We curate a set of competitive CSEGG baselines by directly combining three major categories of continual learning methods with two SGG backbones and benchmark them in our CSEGG dataset. Their inferior performances show the difficulties of our benchmark tasks, which require the ability to expand, modify, retain, and reason scene graphs within the process of adaptive visual scene understanding. Specifically, the weight-regularization methods fail to estimate the importance of learnable parameters given the complicated model design in SGG backbones. Although image-replay methods retain knowledge from prior tasks through replays, the extensive combinatorial complexity of relationships among objects surpasses the complexity accommodated by a restricted set of replay images with efficient storage. Additionally, none of these baseline methods consider the shifts inherent in long-tailed distributions in dynamic scenes.

Here, we present a strategy called "Replays via Analysis by Synthesis", abbreviated as RAS, designed to address the CSEGG challenges. RAS employs scene graphs from previous tasks, breaking them down and re-composing them to generate diverse scene structures. These compositional scene graphs are then used for synthesizing scene images for replays. Due to its nature of symbolic replays, RAS doesn't require the storage of original images, which often carry excessive and redundant details. This also ensures data privacy preservation and data efficiency. Furthermore, by synthesizing scenes using composable scene graphs, RAS maintains the semantic context and structure of previous scenes and also enhances the diversity of scene generation. To prevent biased predictions stemming from long-tailed distributions, we moderate the distribution of replayed scene graphs by balancing tailed and head classes. This ensures a uniform sampling of relationships and objects during replays. Extensive experiments underscore the effectiveness of our approach. Network analysis reveals crucial design choices that can be beneficial for the future development of CSEGG models.

## 2 RELATED WORKS

**Scene Graph Generation Datasets.** Visual Phrase (Sadeghi & Farhadi, 2011) stands as one of the earliest datasets in the field of visual phrase recognition and detection. Over time, various large-scale datasets have emerged to tackle the challenges of Scene Graph Generation (SGG) (Johnson et al.,

2015; Lu et al., 2016; Krishna et al., 2017; Kuznetsova et al., 2020; Liang et al., 2019; Zareian et al., 2020; Yang et al., 2019; Xu et al., 2017; Zhang et al., 2017; Dai et al., 2017; Li et al., 2017b; Zhang et al., 2019a). Among these, the Visual Genome dataset (Krishna et al., 2017) has played a pioneering role by providing rich annotations of objects, attributes, and relationships in images. Despite the significant contributions of these datasets to SGG, none focuses on continual learning in SGG. As the preliminary efforts towards CSEGG, we re-structure the Visual Genome dataset (Krishna et al., 2017) and establish a novel and comprehensive CSEGG benchmark, where AI models are deployed to dynamic scenes where new objects and new relationships are introduced.

**Scene Graph Generation (SGG) Models.** SGG models are categorized into two main approaches: top-down and bottom-up. Top-down approaches(Liao et al., 2019; Yu et al., 2017) typically rely on object detection as a precursor to relationship prediction. They involve detecting objects and then explicitly modeling their relationships using techniques such as rule-based reasoning(Lu et al., 2016) or graph convolutional networks (Yang et al., 2018). On the other hand, bottom-up approaches focus on jointly predicting objects and their relationships in an end-to-end manner (Li et al., 2017a;b; Xu et al., 2017). These methods often employ graph neural networks (Li et al., 2021; Zhang et al., 2019b) or message-passing algorithms (Xu et al., 2017) to capture the contextual information and dependencies between objects. Furthermore, recent works have explored the integration of language priors (Plummer et al., 2017; Lu et al., 2016; Wang et al., 2019) and attention mechanisms in transformers(Andrews et al., 2019) to enhance the accuracy and interpretability of scene graph generation. However, none of these works evaluate SGG models in the context of continual learning. In our work, we directly combine continual learning methods with SGG backbones and benchmark these competitive baselines in CSEGG. Our results reveal the limitations of these methods and highlight the difficulty of our CSEGG learning protocols.

**Continual Learning Methods.** Existing continual learning works can be categorized into several approaches. (1) Regularization-based methods (Kirkpatrick et al., 2017; Chaudhry et al., 2018; Zenke et al., 2017; Aljundi et al., 2018; Benzing, 2022) aim to mitigate catastrophic forgetting by employing regularization techniques in the parameter space. (2) Dynamic architecture-based approaches(Wang et al., 2022a; Yoon et al., 2017; Hung et al., 2019; Ostapenko et al., 2019) adapt the model's architecture dynamically to accommodate new tasks without interfering with the existing ones. (3) Replay-based methods (Rolnick et al., 2019; Chaudhry et al., 2019; Riemer et al., 2018; Vitter, 1985; Rebuffi et al., 2017; Castro et al., 2018) utilize a memory buffer to store and replay past data during training, enabling the model to revisit and learn from previously seen examples, thereby reducing forgetting. The special variants of these methods include generative replay methods, such as (Shin et al., 2017; Wu et al., 2018; Ye & Bors, 2020; Rao et al., 2019), where synthetic data is generated and replayed. Although these generative replay methods, as well as other continual learning methods, have been extensively studied in image classification (Mai et al., 2021; Wang et al., 2022b; Cha et al., 2021) and object detection(Wang et al., 2021b; Shieh et al., 2020; Menezes et al., 2023), few works focus on the problems associated with CSEGG, such as adaptive reasoning from the dynamic scenes, the evolving long-tailed distribution across scenes, and the combinatorial complexity involving objects and their multiple relationships. In this work, we introduce a continual learning strategy, abbreviated as RAS (Replays via Analysis by Synthesis). To address the distinct challenges of CSEGG, RAS involves creating synthetic scenes based on re-composable scene graphs from previous tasks to reinforce continual learning. The components in RAS facilitate memory-efficient training and preserve privacy while maintaining the diversity and context of scene generation in dynamic environments.

## 3 CONTINUAL SCENE GRAPH GENERATION (CSEGG) BENCHMARK

In CSEGG, we re-organize the Visual Genome (Krishna et al., 2017) dataset to cater to the three continual learning scenarios below and follow the standard image splits in (Xu et al., 2017) for training, validation, and test sets. In each learning scenario, we consider a sequence of $T$ tasks consisting of images and corresponding scene graphs with new objects or new relationships, or both. Let $D_t = \{(I_i, G_i)\}_{i=1}^{N_t}$ represent the dataset at task $t$, where $I_i$ denotes the $i$-th image and $G_i$ represents the associated scene graph. The scene graph $G_i$ comprises a set of object nodes $O_i$ and their corresponding relationships $R_i$. Each object node $o_j$ is defined by its class label $c_j$ and its bounding box locations and sizes $b_j$. Each relationship $r_k$ is represented by a triplet $(o_s, p_k, o_o)$, where $o_s$ and $o_o$ denote the subject and object nodes, and $p_k$ represents the relationship predicate.

| Scenarios | #Tasks | #Objs | #Rels | Eval. metrics | SGG Backbone | CL base. |
|-----------|--------|-------|-------|---------------|--------------|----------|
| S1 | 5 | All (150) | Each task :- 10 | F, R, mF, mR, FWT, BWT, R_bbox, Gen_R | Transformer based (SGTR) | Joint, Naive, Replay10%, Replay20%, Replay100%, EWC, PackNet |
| S2 | 2 | Task 1 :- 100 Task 2 :- 25 | Task 1 :- 40 Task 2 :- 5 | | | |
| S3 | 4 | Each task :- 30 | Each task :- 35 | | CNN based (IMP) | |

Table 1: **Overview of CSEGG Learning Scenarios.** This table summarizes the searning scenarios in CSEGG, including the number of tasks, object and relationship classes, SGG-Backbones used, and the continual learning baselines applied in each scenario.

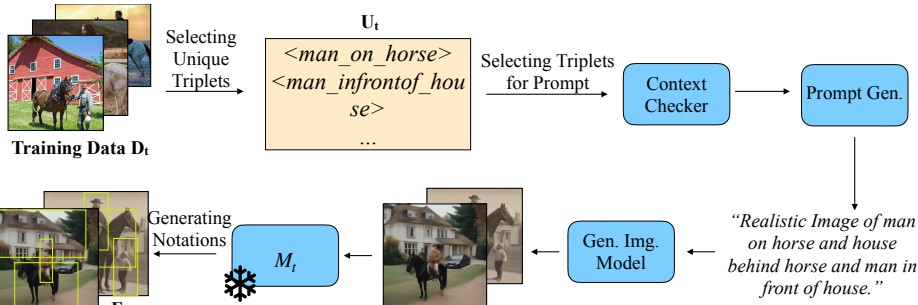

Figure 2: **Overview of RAS method.** $E_{t+1}$ is used as an exempler for training $M_{t+1}$ in task $t_{t+1}$. See **Sec.** 4 for the detailed explanation. See **Sec.** 5.4 and **Tab.** 3 for the ablation results.

## 3.1 LEARNING SCENARIOS

**Scenario 1 (S1): Relationship Incremental Learning.** To uncover contextual information and go beyond studies of object detection and recognition, we introduce this scenario consisting of 5 tasks where 10 new relationship classes are incrementally added in every task (**Fig. S2(S1)**, Tab.1). All object classes and their locations are made known to all CSEGG models over all the tasks. This scenario resembles a human learning scenario where a parent gradually teaches a baby to recognize new relationships among all objects in the same bedroom, focusing on one relationship at a time during continual learning. This scenario also has implications in medical imaging where identical cell types may form new relationships with nearby cells depending on the context. See SecA.1.1 for details.

**Scenario 2 (S2): Scene Incremental Learning.** To simulate the real-world cases when there are demands for detecting new objects and new relationships from old to new scenes, we introduce this scenario where new objects and new relationships are incrementally introduced over tasks (**Fig. S2(S2)**, Tab.1). There are 2 tasks in total with the first task containing 100 object classes and 40 relationship classes with 25 more object classes and 5 more relationship classes in the second task. This aligns with the real-world use cases where common objects and relationships are learned in the first scene, and incremental learning in the second scene only happens on less frequent relationships and objects. See SecA.1.2 for details.

**Scenario 3 (S3): Relationship Generalization.** Humans have no problem at all recognizing the relationships of unknown objects with other nearby objects. This scenario is designed to investigate the relationship generalization ability of CSEGG models. This capability is essential for real-world implications, such as in robotic navigation where it often encounters unknown objects and requires classifying their relationships. There are 4 tasks in total with each task containing 30 object classes and 35 relationship classes (**Fig. S2(S3)**, Tab.1). Different from **S1** and **S2**, a standalone generalization test set is curated, where the objects are unknown but the relationships among these unknown objects are common to the training set of every task. The CSEGG models trained after every task are tested on this standalone test set. See Sec.A.1.3 for details.

**Data sampling and distributions.** To obtain data for every task of each scenario, we perform the following sampling strategies. In scenarios **S1** and **S3** above, either object or relationship classes are randomly sampled from the Visual Genome dataset and incrementally added to every task. Due to the inherent characteristics of real-world scenes, the long-tailed distribution is present. However, in scenario **S2**, only tailed object and relationship classes are sampled and added in subsequent tasks. We showcase the number of images in each task of all scenarios in **Fig. S4,S5,S6** in **Sec. A.3**.

## 3.2 Competitive CSEGG Baselines

Due to the scarcity of CSEGG works, we contribute a diverse set of competitive CSEGG baselines and implement them on our own. Each CSEGG baseline requires three components: a backbone model for scene graph generation (SGG), a continual learning (CL) method to prevent the SGG model from forgetting, and an optional data sampling technique to deal with imbalanced data at every task for training SGG models. Next, we introduce the 2 SGG backbones, the 5 continual learning methods, and the 5 data sampling techniques. For every CSEGG baseline, we follow the naming convention: [backbone] - [CL method] - [sampling technique]; e.g. SGTR-EWC-BLS indicates a CSEGG baseline where the continual learning EWC method is applied on SGTR backbone and BLS sampling technique is used during training at every task in continual learning.

**SGG Backbones.** We use the two state-of-the-art backbones: (1) one-stage Scene graph Generation TRansformer (SGTR) (Li et al., 2022b) and (2) the traditional two-stage SGG model (CNN-SGG) (Xu et al., 2017). Briefly, SGTR (**Fig. S3a**) uses a transformer-based architecture for image feature extraction and fusion and formulates the task as a bipartite graph construction problem. CNN-SGG detects objects with Faster-RCNN(Girshick, 2015) backbone and infers their relationships separately via Iterative message passing (IMP)(Xu et al., 2017). We use public source codes from (Li et al., 2022b) and (Wang et al., 2021b) with default hyperparameters.

**Continual Learning Methods.** We include the following continual learning methods: (1) Naive (lower bound) is trained on each task in sequence without any measures to prevent catastrophic forgetting. (2) EWC(Kirkpatrick et al., 2017) is a weight-regularization method, where the weights of the network are regularized in the parameter space, based on their "importance" to the previous tasks. (3) PackNet(Mallya & Lazebnik, 2018) is a parameter-isolation method, iteratively pruning and pre-training the network parameters so that it can sequentially pack multiple tasks within one network. (4) Replay(Rolnick et al., 2019) includes a memory buffer with the capacity of storing $M$ percentages of images in the entire dataset as well as their corresponding ground truth object and predicate notations depending on the task at each learning scenario. We vary $M = 10\%$, $20\%$, and $100\%$. (5) Joint Training is an upper bound where the SGG model is trained on the entire CSEGG dataset. All experimental results are based on the average over three runs.

**Sampling Methods to Handle Long-Tailed Data.** We adopt the five data sampling techniques to alleviate the problem of imbalanced data distribution during training. (1) LVIS(Gupta et al., 2019) is an image-level over-sampling strategy for the tailed classes. (2) Bi-level sampling (BLS) (Li et al., 2021) balances the trade-off between image-level oversampling for the tailed classes and instance-level under-sampling for the head classes. (3) Equalized Focal Loss (EFL) (Li et al., 2022a) is an effective loss function, re-balancing the loss contribution of head and tail classes independently according to their imbalance degrees. EFL is enabled all the time for all the CSEGG baselines. In addition to applying data sampling techniques to the training sets, we can also apply LVIS and BLS techniques to the data stored in the replay buffer. We name these data sampling techniques during replays as (4) LVIS@Replay and (5) BLS@Replay.

## 3.3 Evaluation Metrics

Same as existing SGG works (Xu et al., 2017; Li et al., 2022b), we adopt the evaluation metric recall@K (**R@K**) on the predicted scene graphs $G$, where $K$ refers to the top-K predictions. As CSEGG is long-tailed, we further report the results in mean recall (**mR@K**) over the head, body, and tail relationship classes. We explored CSEGG with K=20, 50, and 100. Results are consistent among Ks, so we analyze all the results based on K=20 in the entire text. Forgetfullness (F), Average (Avg.) performance, Forward Transfer (FWT) (Lin et al., 2022) and Backward Transfer (BWT) (Lopez-Paz & Ranzato, 2017) are standard evaluation metrics used for continual learning in image recognition and object detection tasks. In Scenario 1 and 2, we adapt these metrics to recalls R@K and introduce F@K, Avg. R@K, FWT@K, and BWT@K respectively for CSEGG settings.

In scenario S3, we evaluate all CSEGG methods in the single standalone generalization test set, shared over all the tasks. To benchmark generalization abilities in unknown object localization and relationship classification among these unknown objects, we introduce two evaluation metrics: the recall of the predicted bounding boxes on unknown objects (**Gen $R_{bbox}$@K**) and the recall of the predicted graph $G_i$ (**Gen R@K**). As the CSEGG models have never been taught to classify unknown objects, we discard the class labels of the bounding boxes and only evaluate the predicted box locations with **Gen $R_{bbox}$@K**. To evaluate whether the predicted box location is correct, we

| Baselines | Learning Scenario 1 (S1) | | | | | | Learning Scenario 2 (S2) | | | | | |
|---|---|---|---|---|---|---|---|---|---|---|---|---|
| | Avg.R ↑ | F ↑ | mR ↑ | mF ↑ | FWT ↑ | BWT ↑ | Avg.R ↑ | F ↑ | mR ↑ | mF ↑ | FWT ↑ | BWT ↑ |
| | SGTR | | | | | | | | | | | |
| Joint | **20.15** | **0** | **4.6** | **0** | - | - | **12.64** | **0** | **9.84** | **0** | - | - |
| Naive | 1.33 | -28.7 | 0.86 | -1.74 | -2.03 | -60.67 | 0.51 | -23.22 | 0.05 | -11.31 | -3.77 | -62.34 |
| Replay 10% | 8.55 | -22.21 | 4.33 | -1.44 | **4.29** | -38.35 | 1.81 | -20.72 | 1.15 | -9.64 | -0.9 | -40.67 |
| Replay 20% | - | - | - | - | - | - | 2.57 | -17.17 | 1.56 | -8.07 | **-0.67** | -38.27 |
| Replay 100% | 16.17 | -12.24 | 3.32 | -1.34 | -1.77 | **-11.72** | 4.56 | -4.13 | 4.56 | -5.61 | -1.045 | **-30.25** |
| EWC | 1.89 | -28.4 | 0.96 | -1.72 | -1.17 | -52.45 | 0 | -23.22 | 0 | -11.31 | -2.65 | -50.12 |
| PackNet | 7.19 | -25.67 | 1.35 | -1.64 | -1.03 | -42.35 | 1.67 | -22.77 | 0.9 | -10.33 | -1.4 | -42.45 |
| Ours* | - | - | - | - | - | - | - | - | - | - | - | - |
| | CNN-SGG | | | | | | | | | | | |
| Joint | **19.53** | **0** | **3.9** | **0** | - | - | **4.3** | **0** | **3.7** | **0** | - | - |
| Naive | 0.98 | -21.2 | 0.74 | -1.35 | -3.45 | -43.87 | 0 | -18.22 | 0.45 | -2.67 | -4.12 | -53.12 |
| Replay 10% | 5.67 | -18.9 | 3.21 | -1.05 | **1.45** | -28.34 | 1.81 | -16.72 | 1.03 | -1.74 | -1.4 | -43.56 |
| Replay 20% | - | - | - | - | - | - | 2.37 | -15.17 | 1.45 | -1.53 | **-1.1** | -38.56 |
| Replay 100% | 13.45 | -8.83 | 3.6 | -0.35 | -1.5 | **-10.45** | 12.45 | -4.13 | 3.2 | -0.56 | -2.1 | **-20.34** |
| EWC | 2.36 | -21.05 | 0.67 | -1.34 | -2.34 | -39.89 | 0 | -18.22 | 0.03 | 0 | -3.77 | -51.67 |
| PackNet | 3.2 | -19.7 | 1.1 | -1.13 | -1.3 | -32.45 | 1.1 | -17.82 | 0.84 | -1.97 | -2.84 | -40.34 |
| Ours* | - | - | - | - | - | - | - | - | - | - | - | - |

Table 2: **Results of average recall and forgetting on last task in Learning Scenarios 1 and 2 based on the SGTR and CNN-SGG backbone as the SGG model.** Here, R.10%, R.20% and R.100% stands for Replay 10%, Replay 20% and Replay 100% respectively. See **Sec. 3.2** for introduction to continual learning baselines. See **Sec. 3.3** for explanations about evaluation metrics. The higher Avg.R and F, the better. * :- Currently we are running experiments on the S1, S2 with RAS. See Sec. 5.4 and Tab. 3 for ablation results.

apply a hard threshold of Intersection over Union (**IoU**) between the predicted bounding box locations and the ground truth. Any predicted bounding boxes with their IoU values above the hard threshold are deemed to be correct. We vary IoU thresholds from 0.3, 0.5, and 0.7. To assess whether the CSEGG model generalizes to detect known relationships over unknown objects, we evaluate the recall **Gen R@K** of the predicted relationships $r_k$ only on *correctly predicted* bounding boxes. See Sec. A.4 for detailed definitions of these metrics. In the main text, we only focus on **K=20.**

## 4 RAS

In our approach, we aggregate a unique set $U$ of triplets from all images $i$ within task $t$. The only information retained in the replay buffer from a task $t$ is set $U$. Unlike the traditional replay method in continual learning literature, our method refrains from storing images $I_i$ or graphs $G_i$. Leveraging generative modeling, we generate exemplar $E_{t+1}$ for the subsequent task using the information stored in this unique set. We get the subsequent model $M_{t+1}$ by training $M_t$ using $D_{t+1}$ and exemplar $E_{t+1}$ to mitigate catastrophic forgetting. This is possible as generated exemplar $E_{t+1}$ contains training examples similar to previous tasks. Next, we explain how we generate $E_{t+1}$ using generative models as shown in **Fig.** 2. To effectively train a model $M$ on an exemplar $E$, the exemplar must consist of pairs $E = \{I_j, G_j\}$, where $I_j$ represents the $j$-th image in the exemplar, and $G_j$ is the corresponding graph of $I_j$, as detailed earlier. Additionally, within $I_j$ and $G_j$ of $E_t$, there is a requirement for training examples akin to the current task $t$ to alleviate catastrophic forgetting in the continual setting. To address this, we create $U_t$, a set of unique triplets present in all images of $\{I_i, G_i\}$ in $D_t$, ensuring that the exemplar $E_t$ encapsulates relevant training instances for the ongoing task.

**Generating Images**. To generate images $I_i$ for $E_t$, we employ state-of-the-art image generation models, specifically leveraging the open-source Stable Diffusion model Rombach et al. (2022). For prompt generation in the image generation model, we strategically select $n$ triplets from $U_t$. To ensure that the generated image is akin to real world settings, we use a context checker. This context checker, generate the embedding vectors for the selected triplets and calculate a similarity score. Only triplets with similarity score $> 0.6$ are used to create prompt for the image generation model. To generate prompt we employ a straightforward English language construct using the conjunction "and". This involves combining all the selected triplets into a sentence starting with "Realistic Image of" serving as a coherent prompt. For instance, if the chosen triplets from $U_t$ are <man, on, horse>, <house, behind, horse>, and <man, in front, house>, the generated prompt becomes "Realistic Image of man on horse and house behind horse and man in front of house." Acknowledging the potential inaccuracies in image generation, we generate multiple images (specifically $a = 10$) from the same

| | Gen. Img. | Multi. Triplets | Gen. GT | Avg.R ↑ | F ↑ |
|---|---|---|---|---|---|
| A1 | ✗ | ✗ | ✗ | 6.09 | -26.75 |
| A2 | ✗ | ✓ | ✗ | **8.55** | **-22.21** |
| A3 | ✗ | ✗ | ✓ | 6.89 | -25.1 |
| A4 | ✗ | ✓ | ✓ | 7.32 | -23.89 |
| A5 | ✓ | ✗ | ✓ | 5.6 | -27.8 |
| A6 | ✓ | ✓ | ✓ | 6.75 | -25.42 |

Table 3: **Ablation study on our RAS on S1 reveals key design insights.** This table presents the results of ablation studies conducted to identify key components of our method, as discussed in **Sec. 5.4**. Avg.R and F are of the last task in S1 scenario.

prompt. This approach ensures that, even if a few generated images exhibit suboptimal quality, the exemplar contains a sufficient number of high-quality images for effective model training.

**Generating Graphs**. To obtain the corresponding $G_i$ for the generated $I_i$ in $E_t$, we utilize the current task model $M_t$ to generate notations. Consequently, all generated $I_i$ from the image generation model are inputted into the current task model $M_t$, producing $G_i$. This $G_i$ comprises object nodes $O_i$ with their respective classes $c_j$, along with object locations $b_j$. Additionally, it includes corresponding relationship nodes $R_i$ formed by triplets $<o_s, p_k, o_j>$ representing subject, predicate, and object nodes, respectively. These generated notations serve to construct the exemplar and are pivotal for future training iterations.

## 5 RESULTS

### 5.1 CONTINUAL SCENE GRAPH GENERATION REMAINS A GREAT CHALLENGE.

We present Avg. R, F, FWT, and BWT results for learning scenario 1 (S1) and learning scenario 2 (S2) in **Tab. 2**. Among all the methods, the naive method takes no measures of preventing catastrophic forgetting, resulting in the largest drop in Avg.R and F. In contrast, a replay method with all the old data to rehearse in the current task (Replay(100%)) yields the least forgetting and maintains a high Avg.R. Surprisingly, even though Replay(100%), as an upper bound, replays all the data in the current and previous tasks, there is still a drop in performance. This could possibly be due to the long-tailed data distribution in the memory buffer, which makes the rehearsal of tail classes even less frequent in new tasks, and thus, deteriorate the recall performances of tail classes. We also compared EWC versus the replay methods. Though EWC outperforms the naive baseline in earlier tasks, it fails in longer task sequences. Different from EWC, Replay with 10% still achieves a higher Avg.R score of 8.55% and a higher F of -22.21%. This aligns with the existing continual learning literature that replay methods are more effective than weight-regularization methods in eliminating catastrophic forgetting (Lesort et al., 2019). PackNet is a parameter isolation method. While PackNet outperforms EWC, its performance is inferior to that of Replay(10%). As expected, we also compare the replay methods with different memory buffer sizes. Replaying more old data helps CSEGG performances. Joint training demonstrates superior performance over all the tasks in Learning Scenario 1 as seen in Fig **Tab. 2**. This aligns with the existing continual learning literature that joint training is a superior upper bound than Replay(100%). As the knowledge carried forward is important for the subsequent tasks, we also permuted the task sequences and explored their role in CSEGG performances. Aligning with the existing literature (Singh et al., 2022), we found a prominent effect of task sequences in CSEGG (**Fig. S16** and **Sec. A.5.4**).

Learning scenario 2 (S2) approximates the real-world CSEGG setting where there are constantly new demands in detecting new objects and new relationships simultaneously. The results of S2 in Avg.R and F are provided in **Tab. 2**. Compared with S1, the overall Avg.R and F drop more significantly over tasks. For example, even with 20% memory buffer size, the replay method only achieves Avg. R@20 score of 2.57% and F@20 of -17.17% in Task 2. This suggests that the real-world CSEGG remains a challenging task and there still exists a large performance gap for state-of-the-art CSEGG methods. Moreover, we also made an interesting observation that Replay (100%) outperforms the upper bound of the joint training in the first task of Scenario 2. This performance difference could be attributed to the presence of long-tailed data distribution across tasks, with the first task containing more tailed classes than head classes. This is in contrast to the task splits in Scenario 1 where both head and tail classes are uniformly sampled for every task. Consequently, joint training struggles in the first task due to sub-optimal performance in tailed classes. To gain a qualitative understanding of CSEGG performances, we provide visualization results of the predicted scene graphs on example

| Model | Avg. R@20 ↑ | | F@20 ↑ | |
|---|---|---|---|---|
| | T1 | T5 | T1 | T5 |
| Replay(10%) | 28.7 | 8.55 | 0 | -22.21 |
| Replay(10%)+LVIS | 28.7 | 14.38 | 0 | -15.39 |
| Replay(10%)+BLS | 28.7 | 9.56 | 0 | -22.4 |
| **Replay(100%)** | **28.7** | **16.17** | **0** | **-12.24** |

Table 4: **Results at Task 1 and 5 in Learning Scenario 1 when sampling techniques on long-tailed distribution are applied.** See **Sec. 3.2** for the introduction to techniques used for long-tailed distributions. The best results are in bold.

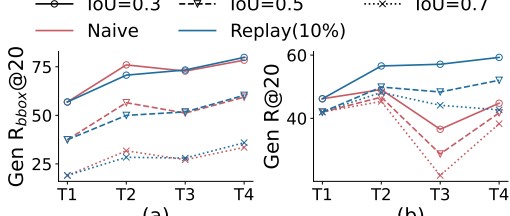

Figure 3: **Generalization results in Learning Scenario 3.** See **Sec. 3.3** for evaluation metrics. The higher the values, the better. Line colors indicate continual learning baselines. Line types denote the IoU thresholds for determining correctly predicted bounding box locations.

images over tasks for all the CSEGG baselines in Scenario 1 (**Fig. S21** and **Sec. A.7.1**) and Scenario 2 (**Fig. S22** and **Sec. A.7.2**).

## 5.2 ANALYZING THE IMPACT OF SAMPLING TECHNIQUES ON LONG-TAILED DISTRIBUTION IN CSEGG

Due to the imbalanced data distribution in the real world, long-tailed distribution remains a unique challenge for CSEGG. Here, we introduce two data sampling techniques (LVIS and BLS) to counter-balance the long-tailed data distributions in the memory buffers as well as the feed-forward training tasks (**Sec. 3.2**). We report the results of Replay(10%)+BLS and Replay(10%)+LVIS in learning scenario S1. In **Tab. 4**, both long-tailed methods with replay(10%) outperform naive replays by an average margin of 3.42% in Avg.R and 3.35% in F. This implies that data re-sampling techniques enhance general continual learning performances in long-tailed incremental learning settings. Indeed, we made the same observations after splitting the classes from each task into tail, body, and head classes and reporting their mR@K in the **Fig. S13** in **A.5.2**. Interestingly, we see that Replay(10%)+BLS underperforms Replay(10%)+LVIS by 4.32% in Avg.R and 6.42% in F. This contradicts the findings that BLS is more effective than LVIS in the classical SGG problem (Li et al., 2021). The performance discrepancy could be due to the difference in the number of replay instances in both approaches after these two data re-sampling methods are applied to the memory buffer (see **Sec. 3.2**). This emphasizes that the long-tailed learning methods explored in the SGG problem may not be effective in CSEGG. We need to explore new long-tailed learning methods specifically for CSEGG.

## 5.3 CSEGG IMPROVES GENERALIZATION IN UNKNOWN SCENE UNDERSTANDING

**Fig. 3** provides the generalization results in detecting unknown objects and classifying known relationships among these objects in Learning Scenario 3 (S3). In **Fig. 3 (a)**, we observed an increasing trend of Gen $R_{bbox}$ for all CSEGG methods as the task number increases. This suggests that CSEGG methods improve generalization abilities in detecting unknown objects, as they learn to continuously detect new objects and classify known relationships among these objects. As expected, with increasing IoU threshold from 0.3 to 0.7, fewer detected bounding boxes are deemed to be correct; thus, there is a decrease in Gen $R_{bbox}$. Subsequently, we observed a decrease in Gen R in relationship generalization in **Fig. 3 (b)** as well. Moreover, we notice that even in Task 1, all CSEGG methods are capable of proposing 23% reasonable object regions with IoU = 0.7. This implies that the SGTR model generalizes to detect "objectness" in the scene even with minimal training only in Task 1. Interestingly, as seen in S1 and S2 (**Tab. 2**), the naive baseline only learns the current task at hand and often forgets the knowledge in old tasks; however, forgetting to detect objects from previous tasks does not interfere its generalization abilities. In fact, its generalization ability to detect unknown objects increases over tasks. Contrary to our previous observations in S1 and S2 (**Sec. 5.1**), where replay methods beat the naive baseline, a surprisingly opposite trend in object detection generalization is observed. One possible explanation is that all CSEGG methods output a fixed number of detected object bounding boxes. As replay methods forget less, they intend to detect more in-domain object boxes out of the total number of bonding boxes they can output, resulting in a decreased number of bounding boxes detected for unknown objects. The results in **Fig. 3 (b)** support this point. Given all the correctly detected unknown object locations, Replay(10%) outperforms the naive baseline. This emphasizes that the continual learning ability to forget less about previous tasks improves the overall generalization abilities of the CSEGG models in unknown scene understanding.

Notably, we also found that the SGTR model is very good at generalizing to classify relationships **Fig. 3 (b)**. Even in Task 1, both the naive method and the Replay(10%) achieve 45% recall of known relationships among unknown objects in the generalization test set. As the CSEGG models continuously learn to detect more new objects and classify their relationships in subsequent tasks, their relationship generalization ability among unknown objects saturates around Task 3. See **Fig. S23** in **Sec. A.7.3** for visualization examples.

## 5.4 ABLATION STUDY ON OUR RAS REVEALS KEY DESIGN INSIGHTS

In order to refine the specifics of our approach, we conducted ablation studies on S1, as outlined in **Table** 3. These studies aimed at informing crucial design choices, including the exploration of diverse image content alongwith determining the optimal strategy for generating images within the exemplar. As depicted in **Table** 3, the performance of images generated from multiple triplets surpassed those from a single triplet which can be seen by comparing ($A1$ and $A2$), then ($A3$ and $A4$) and finally can also be seen in ($A5$ and $A6$), a trend consistently observed across Visual Genome (VG) images and generated images. Another experiment evaluated the quality of notations produced by each task's model by generating notations for VG images ($A4$ in **Table** 3) and incorporating them into the exemplar alongside VG images. As depicted in **Table** 3 by $A2$ and $A4$, applying this approach to VG images resulted in only a minimal drop in Avg.R (16.8%) and F (7%) for the last task of Scenario 1, affirming the quality of the generated notations. Further experimentation involved comparing the performance of generated images with VG images, and as indicated in **Table** 3, the generated images exhibited commendable quality. The comparative analysis in **Table** 3 between $A4$ and $A6$ revealed that when utilizing generated images instead of ground truth images, the drop in Avg.R and F was only 7.32% and 6.01%.

## 6 DISCUSSION

In the dynamic world, the incremental introduction of new objects and new relationships in scenes presents significant challenges for scene graph generation (SGG) models to effectively adapt and generalize without forgetting previously acquired semantic knowledge. However, despite the progress made in SGG and continual learning research, there remains a lack of comprehensive investigations specifically targeting the unique challenges of Continual Scene Graph Generation (CSEGG). To close this research gap, we take the initial steps of operationalizing CSEGG and introducing benchmarks, datasets, and evaluation protocols. Our study delves into three distinct learning scenarios, thoroughly examining the interplay between continual object detection and continual relationship classification for existing CSEGG methods under long-tailed class-incremental settings.

Our experimental results reveal intriguing insights. First, applying standard continual learning approaches combined with long-tailed techniques to SGG models yields moderate improvements. However, a notable performance gap persists between current CSEGG methods and the joint training upper bound. Second, we investigated the model's generalization ability and found that the models are capable of generalizing to classify known relationships involving unfamiliar objects. Third, we compared the CSEGG performance of the traditional CNN-based and the transformer-based SGG models as backbones. We observed consistent relative CSEGG performance across all continual learning methods using both backbones, with CNN-SGG models underperforming SGTR-based ones.

Moving forward, there are several key avenues for future research. Our current endeavors focus on learning CSEGG problems from static images in Independent and Identically Distributed (i.d.d) manner, diverging from human learning from video streams. Future research can look into CSEGG problems on video streams. Our plans also involve expanding continual learning baselines and integrating more long-tailed distribution sampling techniques. Furthermore, we aim to construct a synthetic SGG dataset to systematically quantify the aspects of SGG that influence continual learning performance under controlled conditions. Although the CSEGG method holds promise for many downstream applications like monitoring systems, medical imaging, and autonomous navigation, we should also be aware of its misuse in privacy, data biases, fairness, security concerns, and misinterpretation. We invite the research community to join us in maintaining and updating the safe use of CSEGG benchmarks, thereby fostering its advancements in this field.

## ETHICS STATEMENT

The development and deployment of Scene Graph Generation (SGG) technology present potential negative societal impacts that warrant careful consideration (Li et al., 2022b). Firstly, privacy concerns arise as SGG may inadvertently capture sensitive information from images, potentially violating privacy rights and raising surveillance issues. Secondly, bias and fairness challenges persist, as SGG algorithms can perpetuate biases present in training data, leading to discriminatory outcomes that reinforce societal inequalities. Misinterpretation and misclassification by SGG algorithms could result in misinformation and incorrect actions, impacting decision-making. The risk of manipulation and misuse of SGG-generated scene representations for malicious purposes is also a concern. For example, attackers might manipulate scene graphs to deceive systems or disrupt applications that rely on scene understanding.

## REPRODUCIBILITY STATEMENT

We are committed to ensuring the reproducibility and transparency of our research. In accordance with the guidelines set forth by ICLR 2024, we provide detailed information to facilitate the replication of our experiments and results.

1. **Code Availability:** All code used for our experiments is available at here.
2. **Data Availability:** Any publicly accessible datasets used in our research are specified in the paper, along with their sources and access information.
3. **Experimental Details:** We have documented the specific details of our experiments, including hyper-parameters, model architectures, and pre-processing steps, to enable others to replicate our results.

We are dedicated to supporting the scientific community in replicating and building upon our work. We welcome feedback and collaboration to ensure the robustness and reliability of our research findings.

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

# A Appendix

## A.1 Introduction to Three Learning Scenarios

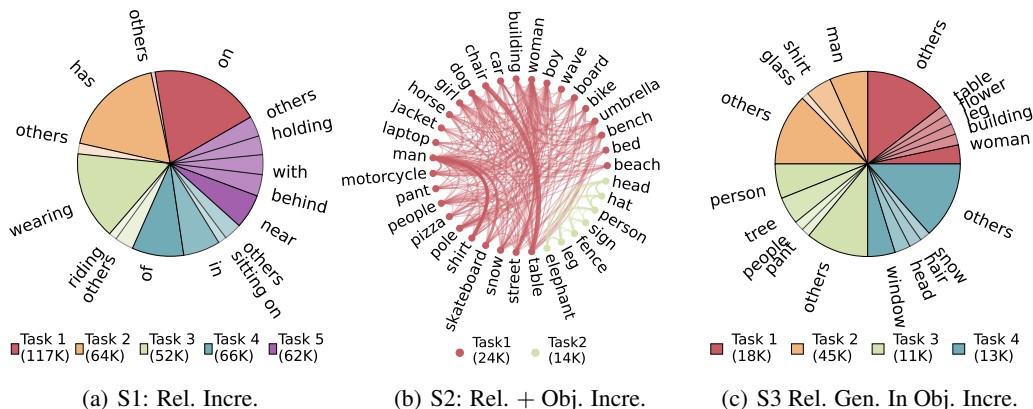

(a) S1: Rel. Incre.  (b) S2: Rel. + Obj. Incre.  (c) S3 Rel. Gen. In Obj. Incre.

Figure S1: **Label distribution in each task in each learning scenario** is presented. In scenario S1 (a) and scenario S3 (c), we use different colors to denote different tasks. The color gradient indicates the frequency of data within a task, with the lighter color denoting the smaller frequency of data in that category. Only the most frequent labels (relationship labels in (a) and object labels in (c)) are provided. See the legend for the total data size per task. In (b) scenario S2 on both objects and relationships, data distributions are presented in the form of small-world networks, where nodes denote object categories and the edges linking object pairs indicate relationships. Thickness in edges implies the diversity of relationships between object pairs. Same color conventions as (a) and (c) are applied. See the legend for triplet sizes. See **Fig. S4,S5,S6** in **Sec. A.3** for the full statistics of S1-3. Within this section, we present more details of three learning scenarios and their practical applications.

### A.1.1 Scenario 1 (S1): Relationship Incremental Learning

While existing continual object detection literature focuses on incrementally learning object attributes (Mai et al., 2021; Wang et al., 2022b; Cha et al., 2021; Wang et al., 2021b; Shieh et al., 2020; Menezes et al., 2023), incremental relationship classifications are equally important as it provides a deeper and more holistic understanding of the interactions and connections between objects within a scene. To uncover contextual information and go beyond studies of object attributes, we introduce this scenario where new relationship predicates $p_k$ are incrementally added in each task (**Fig. S2S1**). There are 5 tasks in S1. To simulate the naturalistic settings where the frequency of relationship distribution is often long-tailed, we randomly and uniformly sample relationship classes from head, body and tail categories in Visual Genome (Krishna et al., 2017), and form a set of 10 relationship classes for each task. Thus, the relationships within a task are long-tailed; and the number of relationships from the head categories of each task is of the same scale. To tackle this issue, we allow CSEGG models to see the same images over tasks, but the relationship labels are only provided in their given task (see **Sec. A.1.5** for the design motivation). The same reasoning applies in **S2** and **S3**. Example relationship classes from each task and their distributions are provided in **Fig. 1(a)**.

Here, we provide a concrete example application of Scenario 1 in medical imaging. Within medical imaging, an agent must acquire the ability to detect cancerous cells within primary tumors, like colon adenocarcinoma. Subsequently, it must extend this proficiency to identifying the same cell types within metastatic growths that manifest in different bodily regions, such as lymph nodes or the liver. In this instance, the identical cancer cell disseminates to fresh organs or tissues, progressively establishing new relationships with other cells over the course of time.

### A.1.2 Scenario 2 (S2): Scene Incremental Learning

To simulate the real-world scenario when there are demands for detecting new objects and new relationships over time in old and new scenes, we introduce this learning scenario where new objects $O_i$ and new relationship predicates $p_k$ are incrementally introduced over tasks (**Fig. S2S2**). To select the object and relationship classes from the original Visual Genome (Krishna et al., 2017) for S2, we

have two design motivations in mind. First, in real-world applications, such as robotic navigation, robots might have already learned common relationships and objects in one environment. Incremental learning only happens on less frequent relationships and objects. (2) Transformer-based AI models typically require large amounts of training data to yield good performances. Training only on a small amount of data from tail classes often leads to close-to-chance performances. Thus, we take the common objects and relationships from the head classes in Visual Genome as one task, while the remaining less frequent objects and relationships from tail classes as the other task. This results in 2 tasks in total with the first task containing 100 object classes and 40 relationship classes. In the subsequent task, the CSEGG models are trained to continuously learn to detect 25 more object classes and 5 more relationship classes. Same as **S1**, both the object class and relationship class distributions are still long-tailed within a task (**Fig. 1(b)**).

Next, we provide two real-world example applications in robot collaborations on construction sites and video surveillance systems.

The CSEGG model's capacity to incorporate new objects and new relationships while retaining existing knowledge finds pivotal application in video surveillance contexts. Consider a company developing video-based security systems for indoor environments, capturing prevalent indoor objects and relationships. Expanding to outdoor settings like parking lots or restricted compounds demands retraining the model with new outdoor data alongside previous indoor data, ensuring operational effectiveness in both realms. The outdoor context introduces new objects like "cars" and relationships like "driving", distinct from indoor scenarios featuring "chair" and "sitting." Employing CSEGG allows the company to focus on new objects and relationships while retaining indoor insights.

Another real-world example would be a construction site where a team of robots is tasked with assembling various components to build a complex structure. Initially, during the foundation-laying phase, the robots are introduced to objects like "concrete blocks" and relationships like "stacking". As the construction advances to the wiring and installation phase, they encounter new objects like "wires" and relationships like "connecting," which were absent from earlier stages. The SGG model deployed in these robots needs to adapt incrementally to learn these new relationships without forgetting the existing ones. This ensures that the robots can effectively communicate and collaborate while comprehending the evolving scene and tasks, optimizing their construction efficiency and accuracy.

### A.1.3 SCENARIO 3 (S3): SCENE GRAPH GENERALIZATION IN OBJECT INCREMENTAL LEARNING

We, as humans, have no problem at all recognizing the relationships of unknown objects with other nearby objects, even though we do not know the class labels of the unknown objects. This scenario is designed to investigate whether the CSEGG models can generalize as well as humans. Specifically, there are 4 tasks in total with each task containing 30 object classes and 35 relationship classes. In each subsequent task, the CSEGG models are trained to continuously learn to detect 30 more object classes and learn to classify the same set of 35 relationships among these objects. The class selection criteria for each task follow the same as **S1**, where the selections occur uniformly over head, body, and tail classes. Example object classes and their label distributions for each task are provided in **Fig. 1(c)**. Different from **S1** and **S2**, a standalone generalization test set is curated, where the objects are unknown and their classes do not overlap with any object classes in the training set but the relationships among these unknown objects are common to the training set of every task. The CSEGG models trained after every task are tested on the same generalization test sets.

Here, we provide two real-world applications of Scenario 3 in the deep sea and space explorations for autonomous navigation systems.

A prime example is the ongoing research on deep sea exploration for autonomous navigation systems, where undiscovered flora and fauna reside beneath the ocean's surface. Encountering new and unidentified species becomes manageable through SGG's ability to understand spatial relations. The robot discerns the object's proximity or orientation even without precise identification of the spieces, enhancing its autonomous navigation ability. Likewise, in deep space exploration, SGG aids in recognizing spatial relationships with previously unseen space debris, aiding in path-planning. In essence, SGG's relationship generalization empowers robots to navigate and plan routes in unfamiliar terrains, such as deep sea and deep space, where novel encounters demand adaptable responses.

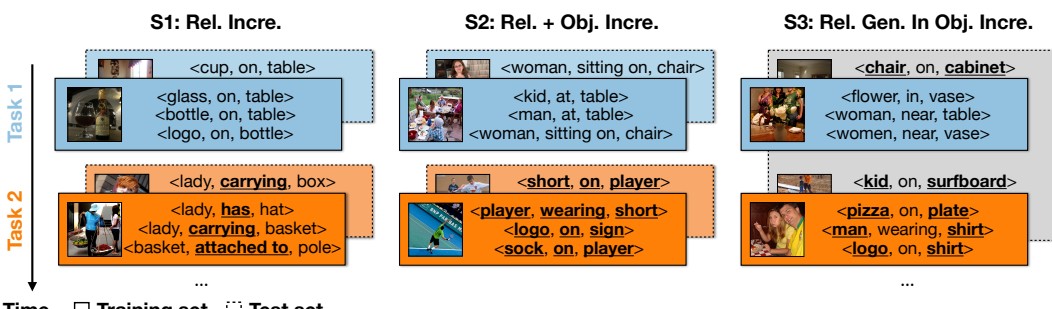

Figure S2: **Three learning scenarios** are introduced. From left to right, they are S1. relationship (Rel.) incremental learning (Incre.); S2. relationship and object (Rel. + Obj.) Incre.; and S3. relationship generalization (Rel. Gen.) in Object Incre.. In S1 and S2, example triplets in the training (solid line) and test sets (dash line) from each task are presented. The training and test sets from the same task are color-coded. The new objects or relationships in each task are bold and underlined. In S3, one single test set (dashed gray box) is used for benchmarking the relationship generalization ability of object incre. learning models across all the tasks.

### A.1.4 PROFOUND IMPACTS BEYOND THE PROBLEM OF CSEGG

We wish to emphasize that the challenge of continuous Scene Graph Generation (SGG) applied to real-world images exhibits two distinctive attributes. Techniques developed to address these attributes possess the potential to be widely applicable across various domains and subsequent tasks. Firstly, SGG in a continuous learning context frequently deals with data stemming from distributions that exhibit long-tailed characteristics. The methodologies formulated within the realm of Continual SGG can be adapted to address broader long-tailed continuous learning issues, such as classifying bird sounds in ecological studies. Secondly, the process of continual SGG demands the ability to engage in reasoning and integrate knowledge over time. As an illustration, in the context of retail inventory management, merely learning to identify durians (a tropical fruit uncommon in US supermarkets) is insufficient. The model must also cultivate the capability to integrate this new information into its existing knowledge database, ensuring that this new product is positioned alongside other fruits within the grocery store.

### A.1.5 DESIGN MOTIVATIONS FOR OVERLAPPING IMAGES AND NON-OVERLAPPING LABELS ACROSS TASKS

The design of such images and label splits over tasks aligns with human learning scenarios where a parent teaches the baby to recognize different toys and objects in the bedroom. Though the baby is exposed to the same bedroom scenes multiple times, the parent only teaches the baby to detect and recognize one object at a time in a continual learning setting. In the future, we will expand our studies to cases where the SGG models learn from non-overlapping sets of training images for each task.

## A.2 IMPLEMENTATION DETAILS

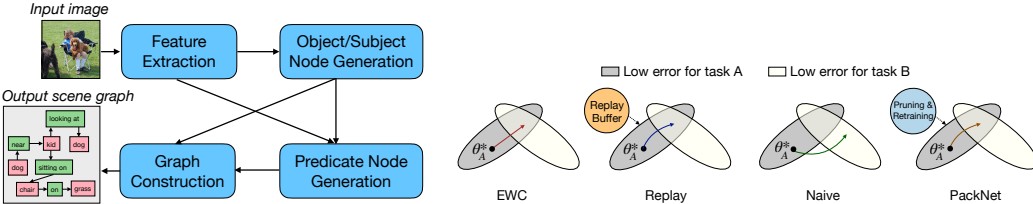

(a) Schematic of SGTR (Li et al., 2022b)          (b) Overview of three continual learning baselines

Figure S3: **Introduction to backbone SGG models and continual learning baselines.** We use Scene graph Generation TRansformer (SGTR) (Li et al., 2022b) as the backbone SGG model (**Sec. 3.2**). SGTR consists of four modules indicated by each blue box. Arrows indicate the signal flows among modules. (b) Four continual learning baselines are listed: EWC (Kirkpatrick et al., 2017), Replay (Rolnick et al., 2019), Naive (**Sec. 3.2**) and PackNet (Mallya & Lazebnik, 2018)(**Sec. 3.2**). $\theta_A^*$ denotes the optimal network parameters after learning on task A. The arrows in colors indicate the shifts of network weights in the parameter space when learning Task B for different baselines.

### A.2.1 SGTR

For SGTR as depicted in (**Fig. S3a**), the approach uniquely formulates the task as a bipartite graph construction problem. Starting with a scene image ($I_i$), SGTR utilizes a 2D-CNN and transformer-based encoder to extract image features. These features are then incorporated into a transformer-based decoder, predicting object and subject nodes ($O_i$). Predicate nodes ($R_i$) are formed based on both image features and object node features, and a bipartite graph ($G_i$) is constructed to represent the scene collectively. The correspondence between object nodes ($o_i$) and predicate nodes ($r_k$) is established using the Hungarian matching algorithm (Kuhn, 1955). Experimental results are based on the average over three runs, and the implementation leverages public source codes from (Li et al., 2022b) and (Wang et al., 2021b) with default hyperparameters. Refer to **Sec. A.2.1** for detailed training and implementation specifics.

The SGTR is trained in two stages in a supervised manner. In stage 1, only object detection losses in DETR is (Carion et al., 2020) applied on $O_i$. In stage 2, only predicate entity prediction loss is applied on $R_i$, which can be further decomposed into L1 and GIOU losses for object/subject/predicate localization (Rezatofighi et al., 2019) and cross-entropy loss for object/subject/predicate classification. In learning scenario S1, we skip Stage 1, and directly load pre-trained weights of DETR for object detection on the entire training set of Visual Genome (Li et al., 2022b). In stage 2 of S1, we freeze the feature extractor, and fine-tune the rest parts of SGTR for predicate entity predictions. As only relationship classes are incrementally introduced in S1, we freeze the entire weights of DETR for detecting all the objects in the scene over tasks. However, empirical results suggest that fine-tuning transformer-based encoders in DETR helps downstream predicate predictions (Li et al., 2022b). Even with fine-tuning DETR in S1, we verify that there is minimal forgetting of detecting all objects in the scene over tasks (see **Sec. S17** and **Sec. A.5.5**). Thus, the forgetting observed in S1 could only be attributed to incremental relationship learning. In Stage 1 of S2 and S3 where object classes are also incrementally introduced over tasks, we load weights of the feature extractor, pre-trained on ImageNet (Deng et al., 2009), and fine-tune the entire DETR (Carion et al., 2020) over all the tasks. Stage 2 of S2 and S3 is the same as S1.

Training the SGTR model involves two stages:

**Object Detection Training:** In this stage, a batch size of 32 is used. All methods are optimized using the Adam optimizer with a base learning rate of $1 \times 10^{-4}$ and a weight decay of $1 \times 10^{-4}$. Object detection training is conducted only in the S2 and S3 scenarios. Each task in S2 is trained for 100 epochs, while each task in S3 is trained for 50 epochs. To expedite convergence, pre-trained weights on ImageNet are utilized before training on Task 1 for both S2 and S3.

**SGG (Scene Graph Generation) Training:** In this stage, the entire SGTR model is fine-tuned while keeping the 2D-CNN feature extractor frozen. A batch size of 24 is employed, and the Adam optimizer is used with a base learning rate of $8 \times 10^{-5}$. In S1 and S3, each model is trained for 50 epochs per task, while in S2, 80 epochs per task are used. All models are trained on 4 A5000 GPUs.

### A.2.2 CNN-SGG BACKBONE

As for CNN-SGG, it employs Faster-RCNN (Girshick, 2015) to generate object proposals from a scene image ($I_k$). The model extracts visual features for nodes and edges from these proposals, and through message passing, both edge and node GRUs output a structured scene graph. Experimental results are based on the average over three runs. Refer to **Sec. A.2.2** for detailed training and implementation specifics.

Given a scene image $I_i$, CNN-SGG utilizes Faster-RCNN(Girshick, 2015) to generate a set of object proposals. The model subsequently extracts visual features of nodes and edges from the set of object proposals. Finally, both edge and node GRUs output a structured scene graph via message passing. The CNN-SGG is trained in two stages in a supervised manner. In stage 1, only object detection losses in Faster-RCNN(Ren et al., 2015) are applied on $O_i$. We use the cross entropy loss for the object class and $L1$ loss for the bounding box offsets. In stage 2, the visual feature extractor (VGG-16(Simonyan & Zisserman, 2014) pre-trained on ImageNet (Deng et al., 2009)) and GRUs layers are trained to predict the final object classes, bounding boxes, and relationship predicates using cross-entropy loss and $L1$ loss. In Learning Scenario 1 (S1), similar to the implementation details of SGTR in **Sec.??**, we skip Stage 1, and directly load pre-trained weights of Faster-RCNN for object detection on the entire training set of Visual Genome (Li et al., 2022b). In stage 2 of S1, we load the pre-trained weights of the visual feature extractor (pre-trained on ImageNet) and fine-tune the rest parts of the model. In stage 2 of S1, we load the pre-trained weights of visual feature extractor (pre-trained on ImageNet) and fine-tune the rest parts of the model. In Stage 1 of S2 and S3 where object classes are also incrementally introduced over tasks, we load weights of the Faster-RCNN, pre-trained on ImageNet Deng et al. (2009), and fine-tune it over all the tasks. Stage 2 of S2 and S3 follows the same training regimes as Stage 2 of S1.

**Object Detection Training:** All methods are optimized using the SGD optimizer with a base learning rate of $1 \times 10^{-2}$ and a weight decay of $1 \times 10^{-4}$. For training on the entire VG dataset, we train the model for 60 epochs with a batch size of 8 for both S2 and S3. To expedite convergence, pre-trained weights on ImageNet are utilized before training on Task 1 for both S2 and S3.

**SGG (Scene Graph Generation) Training:** A batch size of 12 is employed, and the SGD optimizer is used with a base learning rate of $1 \times 10^{-2}$ and a weight decay of $1 \times 10^{-4}$. In S1, each model is trained for 30 epochs. In S2, each model is trained for 15 epochs. In S3, each model is trained for 25 epochs. All models are trained on 4 A5000 GPUs.

### A.2.3 DEFINITIONS OF BWT AND FWT EVALUATION METRICS

To assess the influence that learning a task $t$ has on the performance of any previous tasks in CSEGG models, we also report **Backward Transfer** (**BWT@K**)(Lopez-Paz & Ranzato, 2017). BWT@K is defined as $BWT@K = \frac{1}{T-1} \sum_{i=1}^{T-1} R@K_{T,i} - R@K_{i,i}$, where $T$ denotes the total number of tasks in a learning scenario and $R@K_{i,j}$ denotes the continual learning model trained after task $i$ and tested in task $j$.

To assess the influence of previous tasks on the current task $t$ in CSEGG models, we also report **Forward Transfer** (**FWT@K**) (Lin et al., 2022). FWT is defined as $FWT@K = \frac{1}{T-1} \sum_{i=2}^{T} R@K_{i,i} - \overline{b@K}_{i,i}$, where $\overline{b@K}_{i,i}$ is the **R@K** for an independent model with random initialization trained in task $i$ and tested in task $i$.

A.3 DATA STATISTICS

In this section, we provide various types of data statistics for all three learning scenarios. Specifically, we present statistics regarding the number of images, objects, and relationships involved in each task of each learning scenario. Additionally, we include data statistics pertaining to replay buffers of sizes 10%, 20%, and 100%.

### A.3.1 SCENARIO 1 (S1): RELATIONSHIP INCREMENTAL LEARNING

In this scenario, new relationship predicates $p_k$ are incrementally added in each task. The Learning Scenario 1 (S1) comprises five tasks, with each task consisting of 10 mutually exclusive relationships. Across all tasks, there is a common set of 150 objects present. **Fig. S4** shows comprehensive data statistics for stages 1 and 2 over all the tasks.

As mentioned in **Sec. A.2**, we skip the Stage 1 training in S1 and directly load pre-trained weights of DETR (Zhu et al. (2020)) for object detection on the entire training set of Visual Genome. **Fig. S4(a)** shows the distribution of objects in the entire training set of Visual Genome.

**Fig. S4(b)** displays the distribution of relationships during Training Stage 2 for each task in the Learning Scenario 1 (S1). Each task consists of 10 relationships that are mutually exclusive. Notably, a long tail pattern is observed in the distribution of relationships for each task. The legend of **Fig. S4(b)** indicates that there is a relatively uniform number of images present for each task in Stage 2 of the Learning Scenario 1 (S1).

The test sets for each task in S1 exhibit the distributions of the number of images and relationships that closely align with the distributions of the training sets depicted in **Fig. S4(b)**.

### A.3.2 SCENARIO 2 (S2): SCENE INCREMENTAL LEARNING

In this learning scenario, new objects $O_i$ and new relationship predicates $p_k$ are incrementally introduced over tasks. The Learning Scenario 2 (S2) comprises of 2 tasks with the first task containing 100 objects and 40 relationships and the second task containing 25 objects and 5 relationships. **Fig. S5** shows comprehensive data statistics pertaining to Learning Scenario 2 (S2).

As mentioned in **Sec. A.2**, both the Stage 1 and Stage 2 training are present in S2. **Fig. S5(a)** presents the distribution of objects during Stage 1 for each task in the S2. The first task consists of 100 objects and the second task consists of 25 objects which are mutually exclusive. Notably, a long tail pattern is observed in the distribution of objects for each task. The legend of **Fig. S5(a)** indicates that there is a uniform number of images present for each task in Stage 1 of the S2.

**Fig. S5(b)** displays the distribution of relationships during Stage 2 for each task in the S2. The first task consists of 40 relationships and the second task consists of 5 relationships that are mutually exclusive. Notably, a long tail pattern is observed in the distribution of relationships for each task. The legend of **Fig. S5(b)** highlights that in Stage 2 of the S2, Task 1 has a noticeably higher number of images compared to Task 2. However, the number of relationship notations per class from Task 2 is higher than Task 1, since relationships from head classes are assigned to Task 2 (see **Sec. 3.1**).

The test sets for each task in S2 exhibit similar distributions in the number of images, objects, and relationships to the distributions of the training sets depicted in **Fig. S5(a)(b)**.

### A.3.3 SCENARIO 3 (S3): SCENE GRAPH GENERALIZATION IN OBJECT INCREMENTAL LEARNING

In this learning scenario, there are a total of four tasks, with each task encompassing 30 distinct objects and 35 common relationships. **Fig. S6** shows comprehensive data statistics pertaining to Learning Scenario 3 (S3).

As mentioned in **Sec. A.2**, both Stage 1 and Stage 2 training are present in S3. **Fig. S6(a)** presents the distribution of objects during Stage 1 for each task in the S3. Each task consists of 30 objects which are mutually exclusive. Notably, a long tail pattern is observed in the distribution of objects for each task. The legend of **Fig. S6(a)** indicates that there is a relatively uniform number of images present for each task in Stage 1 of the S3. **Fig. S6(b)** displays the distribution of relationships during

Stage 2 for each task in the S3. Each task consists of 35 relationships which are common over all the tasks. Notably, a long tail pattern is observed in the distribution of relationships for each task.

As mentioned in **Sec. 3.1** of Scenario 3 (S3), a standalone generalization test set is curated for S3. This test set consists of 1942 images and features a completely different set of objects compared to those present in the training data. However, the test set maintains the same set of relationships as observed in the training data.

### A.3.4 REPLAY BUFFERS

In our benchmarking process, we evaluate multiple replay baselines using a memory buffer with a fixed capacity to store a certain percentage (M) of images from the entire training dataset, along with their corresponding ground truth object and relationship notations specific to each learning scenario. We vary the value of M to be 10%, 20%, and 100%. As there could be multiple ground truth objects and relationships per image, we provide statistics on the number of ground truth notations stored in the memory buffer for each task in all the learning scenarios in this section. Specifically, we report the histograms for ground truth notations in the replay buffer for scenario 1 (**Fig. S4**), scenario 2 (**Fig. S5**), and scenario 3 (**Fig. S6**).

In general, over all three learning scenarios, we observe a long-tail distribution of notations per object or relationship class within each task. At stage 1, as we fix the memory capacity over tasks, the number of images stored in the memory buffer remains constant over tasks in most cases, except for memory capacity $M = 100\%$. This is to maximize the diversity of information for replays. Note that although the number of images remain constant in the memory buffer, the number of images allocated to each task decreases given a fixed memory buffer capacity; and hence, we saw a decrease in the number of notations from previous tasks.

At stage 2, the images stored in stage 1 are carried over. However, as some images might not contain task-relevant relationship classes, the number of images used for training at stage 2 might vary over tasks. The notations stored in the memory buffer also follow the long-tail distributions.

In learning scenario 1, we reported the histograms of ground truth notations after two sampling techniques, LVIS and BLS, are applied in the memory buffers (**Fig. S7**). Both LVIS and BLS resampling methods aim to address the long-tail distribution issues by either reducing the prominence of head classes or over-sampling the object instances from the tail classes. Indeed, by comparing the histograms in **Fig. S7(a)(b)**, we noticed such changes in histograms for tail and head classes. A similar trend in the class distribution per task emerges when applying long-tail distribution techniques to the exemplar dataset, as indicated in **Fig.S7(c)(d)**.

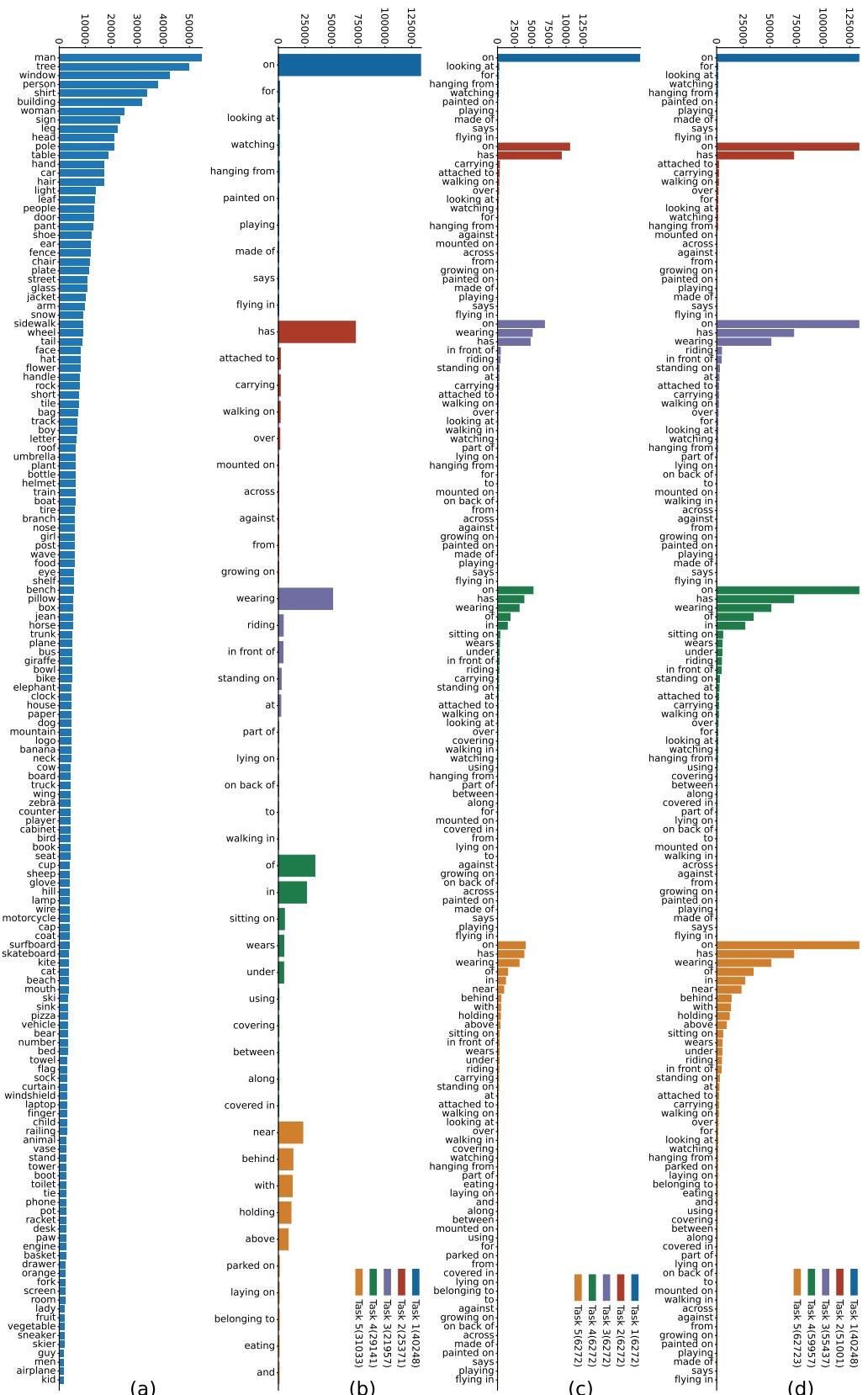

Figure S4: **Data Statistics for Learning Scenario 1 (S1)**: **(a)** Distribution of objects in the entire training set of Visual Genome during Stage 1. **(b)** Distribution of relationships during Stage 2 for each task in S1. **(c)** and **(d)** show distributions of relationships in the memory buffer for Replay 10% and Replay 100 % during Stage 2 for each task in S1. See the legend for the color codes of each task. The numbers in brackets in the legend in **(b-d)** denote the number of training images in the particular task. Zoom in to the figure to get the exact labels and the frequency associated with them.

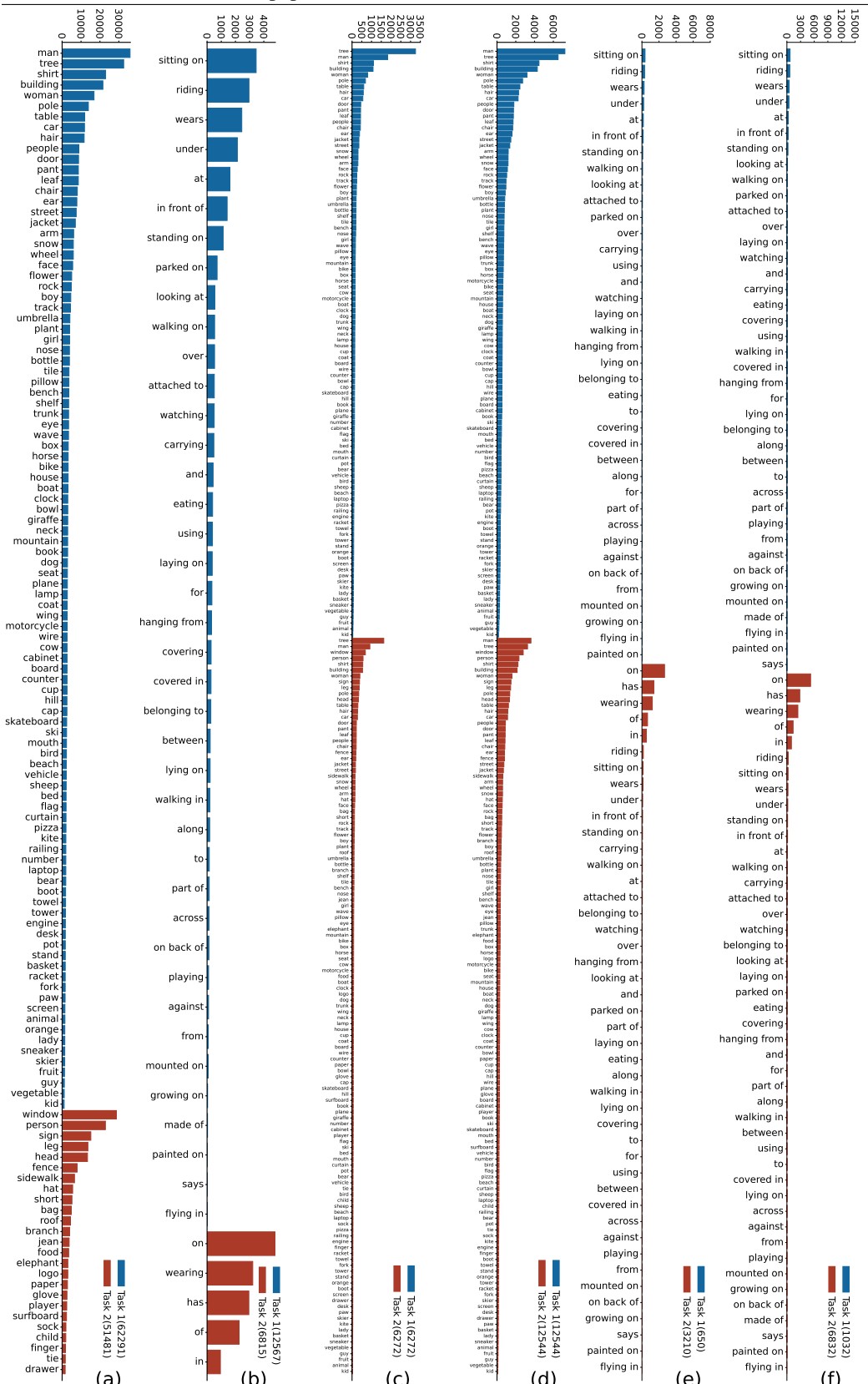

Figure S5: **Data Statistics for Learning Scenario 2 (S2)**: **(a)** Distribution of objects during Stage 1 for each task in S2. **(b)** Distribution of relationships during Stage 2 for each task in S2. **(c)** and **(d)** show distributions of objects in the memory buffer for Replay 10% and Replay 20% during Stage 1 for each task in S2. **(e)** and **(f)** show distributions of relationships in the memory buffer for Replay 10% and Replay 20% during Stage 2 for each task in S2. See the legend for the color codes of each task. The numbers in brackets in the legend in **(a-f)** denote the number of training images in the particular task. Zoom in to the figure to get the exact labels and the frequency associated with them.

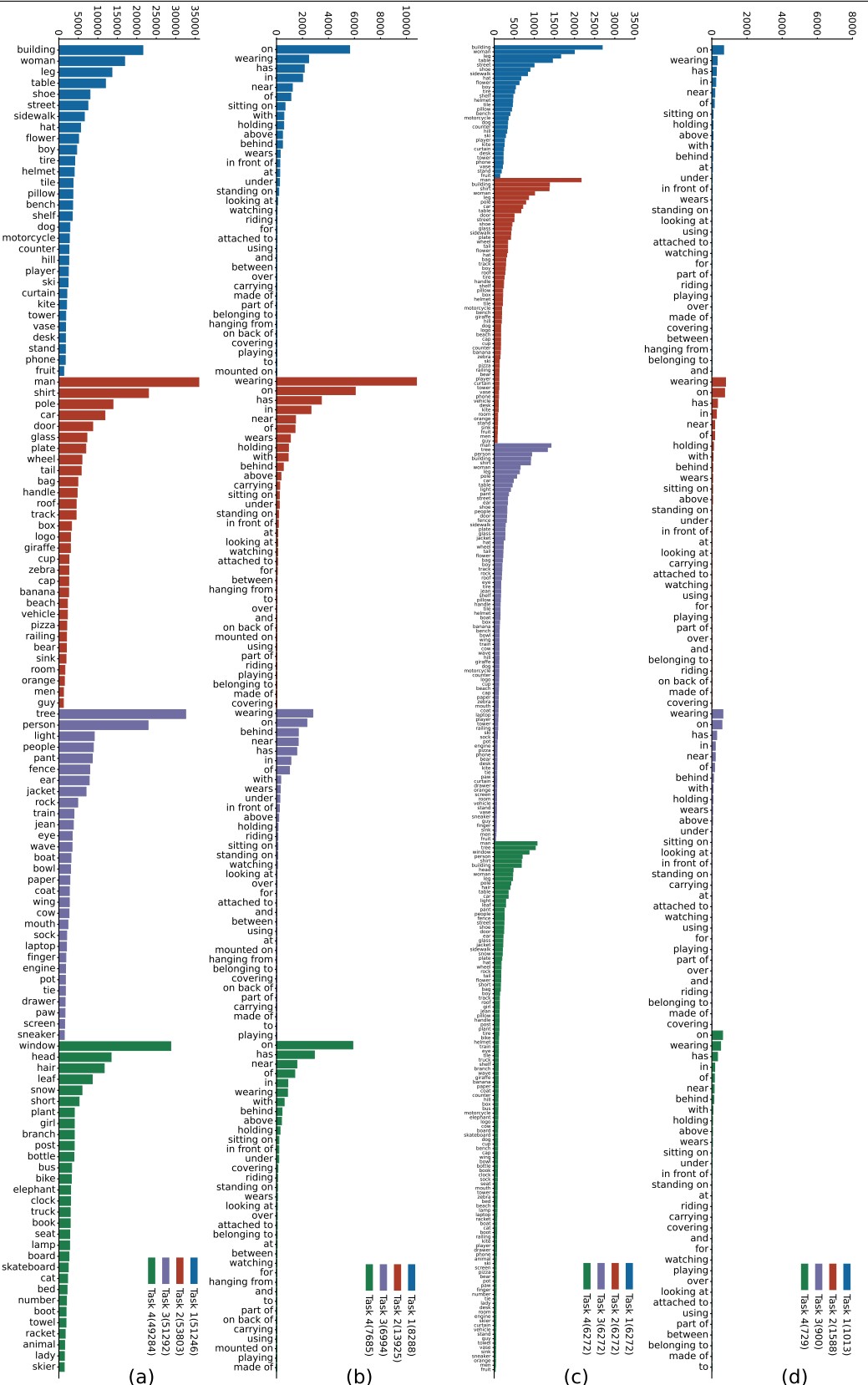

Figure S6: **Data Statistics for Learning Scenario 3 (S3)**: **(a)** Distribution of objects during Stage 1 for each task in S3. **(b)** Distribution of relationships during Stage 2 for each task in S3. **(c)** Distribution of objects in the memory buffer for Replay 10% during Stage 1 for each task in S3. **(d)** Distribution of relationships in the memory buffer for Replay 10% during Stage 2 for each task in S3. See the legend for the color codes of each task. The numbers in brackets in the legend in **(a-d)** denote the number of training images in the particular task. Zoom in to the figure to get the exact labels and the frequency associated with them.

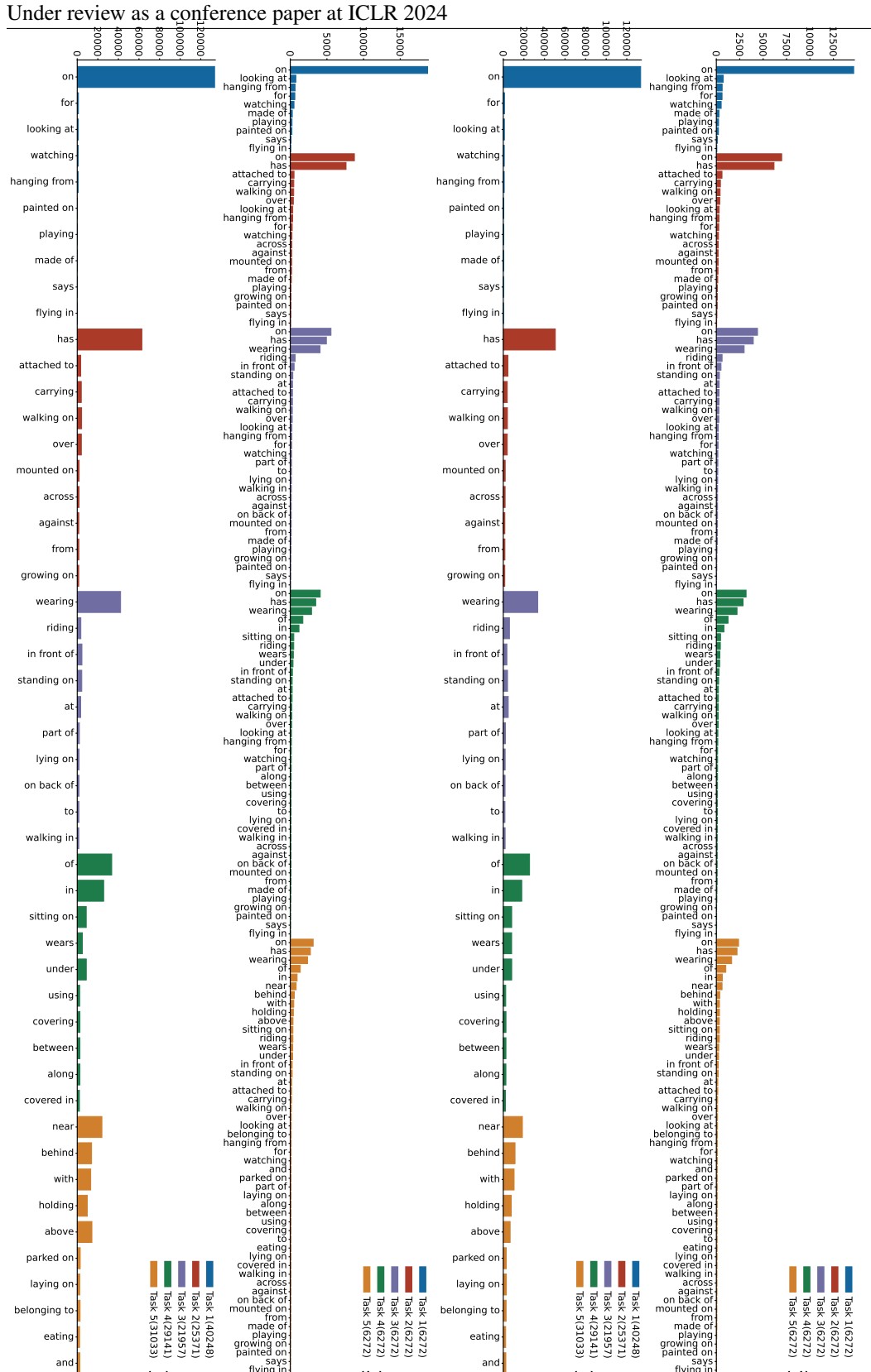

Figure S7: **Data Statistics for Learning Scenario 1 (S1) when sampling techniques on long-tailed distribution are applied**: **(a)** Distribution of relationships during Stage 2 for each task in S1 when LVIS is applied. **(b)** Distribution of relationships in the memory buffer for Replay 10% during Stage 2 for each task in S1 when LVIS is applied. **(c)** Distribution of relationships during Stage 2 for each task in S1 when BLS is applied. **(d)** Distribution of relationships in the memory buffer for Replay 10% during Stage 2 for each task in S1 when BLS is applied. The numbers in brackets in the legend in **(a-d)** denote the number of training images in the particular task. Zoom in to the figure to get the exact labels and the frequency associated with them.

## A.4 EVALUATION METRICS

To assess the catastrophic forgetting of CSEGG models, we define **Forgetfullness (F@K)** as the difference in R@K on $D_{t=1}$ between the CSEGG models trained at task $t$ and task 1. An ideal CSEGG model could maintain the same $R@K$ on $D_{t=1}$ over tasks; thus, $F = 0$ for all tasks. The more negative F is, the more severe in forgetting an model gets. To assess the overall recall of CSEGG models over tasks, we also report the continual average recall (**Avg. R@K**). Avg. R@K is computed as the average recall on all the data at the previous and current tasks $D_i$, where $i \in \{1, 2, ..., t\}$.

To assess whether the knowledge at previous tasks facilitates learning the new task and whether the knowledge at new tasks enhances the performances at older tasks, we introduce **Forward Transfer (FWT@K)** (Lin et al., 2022) and **Backward Transfer (BWT@K)**(Lopez-Paz & Ranzato, 2017). See **Sec. A.2.3** and **Sec. A.5.3** for its definitions, results, and analysis.

In learning scenario S3, we evaluate CSEGG models on their abilities to generalize to detect unknown objects and classify known relationships on these objects, in the standalone generalization test set over all tasks. To benchmark these, we introduce two evaluation metrics: the recall of the predicted bounding boxes on unknown objects (**Gen $R_{bbox}$@K**) and the recall of the predicted graph $G_i$ (**Gen R@K**). As the CSEGG models have never been taught to classify unknown objects, we discard the class labels of the bounding boxes and only evaluate the predicted box locations with **Gen $R_{bbox}$@K**. To evaluate whether the predicted box location is correct, we apply a hard threshold of Intersection over Union (**IoU**) between the predicted bounding box locations and the ground truth. Any predicted bounding boxes with their IoU values above the hard threshold are deemed to be correct. We vary IoU thresholds from 0.3, 0.5, and 0.7. To assess whether the CSEGG model generalizes to detect known relationships over unknown objects, we evaluate the recall **Gen R@K** of the predicted relationships $r_k$ only on *correctly predicted* bounding boxes. For simplicity and consistency, we report the results of **Avg.R@20** and **F@20**. See **Sec. A.5.1** for results at $K = 50, 100$. In general, the conclusions at $K = 50, 100$ are consistent with the cases when $K = 20$.

### A.5 MORE QUANTITATIVE RESULT ANALYSIS ON CONTINUAL SGTR METHODS

#### A.5.1 RESULTS FOR K = 50, 100

**Fig. S8** and **S9** present the results for Avg.R@K and F@K with K values of 50 and 100, focusing on the Learning Scenarios 1 and 2. These results align with the findings discussed in **Sec. 5.1**, which considered Avg.R@20 and F@20. Notably, when K is increased to 50 and 100, both Avg.R and F exhibit higher absolute values compared to K=20 but the trends remain consistent. **Fig. S10** presents the results of Gen.R@K for K=50,100 for Learning Scenario 3. Similar findings in **Sec. 5.3** can be made here as well.

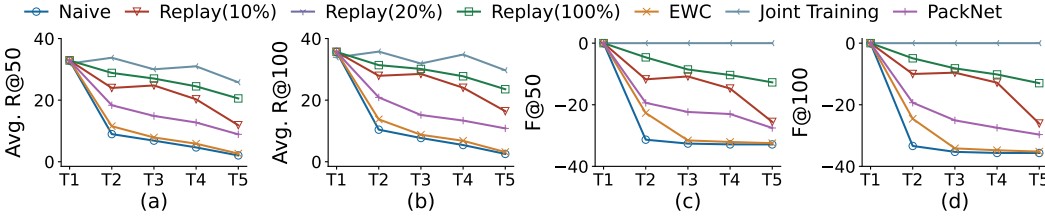

Figure S8: **Results in AvgR@K and F@K over tasks in Learning Scenario 1 (S1) when K=50,100**.

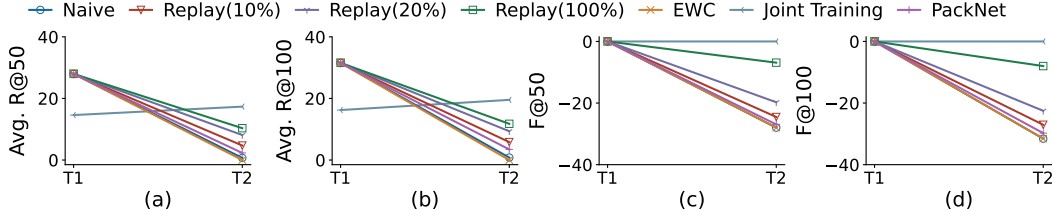

Figure S9: **Results in AvgR@K and F@K over tasks in Learning Scenario 2 (S2) when K=50,100**.

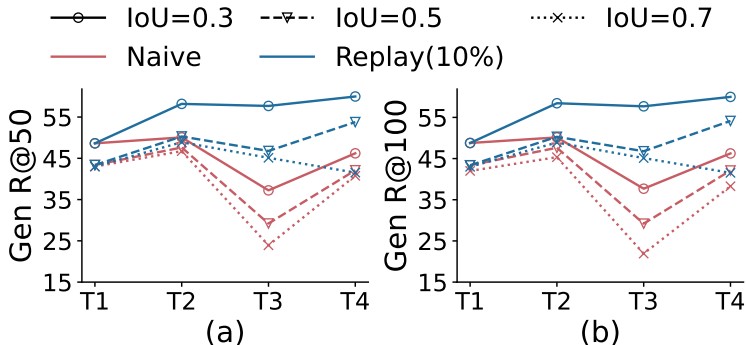

Figure S10: **Results in Gen R@K over tasks in Learning Scenario 3 (S3) when K=50,100**. See **Fig. 3** for design conventions.

### A.5.2 MEAN RECALL RESULTS

Mean Recall@K (mR@K) calculates Recall@K for each relationship category independently and then reports their mean. This is a more fair evaluation metric when dealing with long tail distributions as it averages out the recall over all the relationship classes. In this section, we define mF@K and Avg.mR@K based on mR as described in **Sec. 3.3**. We report the results of mF@K and Avg.mR@K for learning scenarios 1 and 2 in **Fig. S11** and **Fig. S12**. Note that the evaluations with mF@K and Avg.mR@K are not applicable in learning scenario 3. Though the absolute values of mF@K and Avg.mR@K are generally smaller than the values of F@K and Avg.R@K in **Tab. 2**, the same observations made in **Sec. 5.1** can be applied here as well.

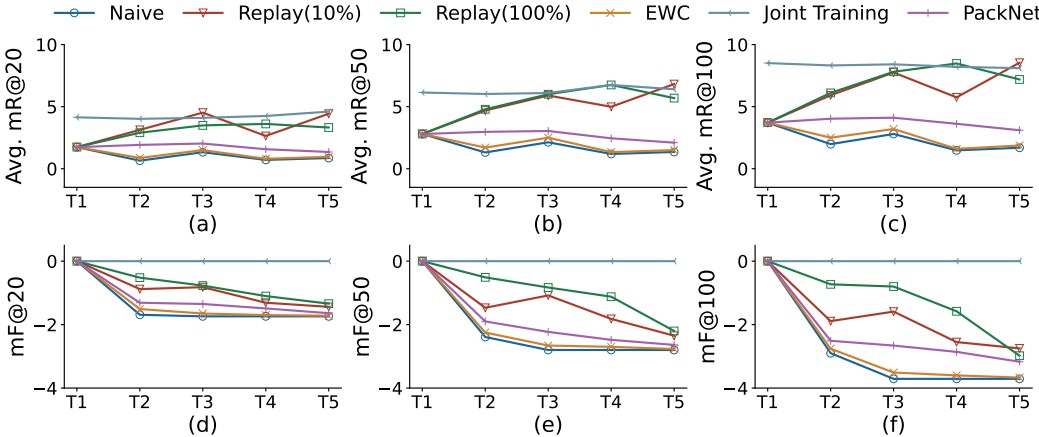

Figure S11: **Results in Avg. mR@K and mF@K over tasks in Learning Scenario 1 (S1) when K=20,50,100**.

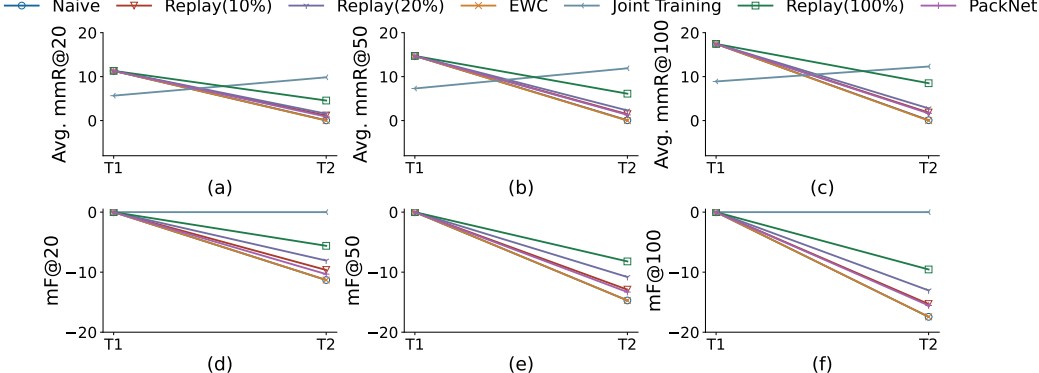

Figure S12: **Results in Avg. mR@K and mF@K over tasks in Learning Scenario 2 (S2) when K=20,50,100**.

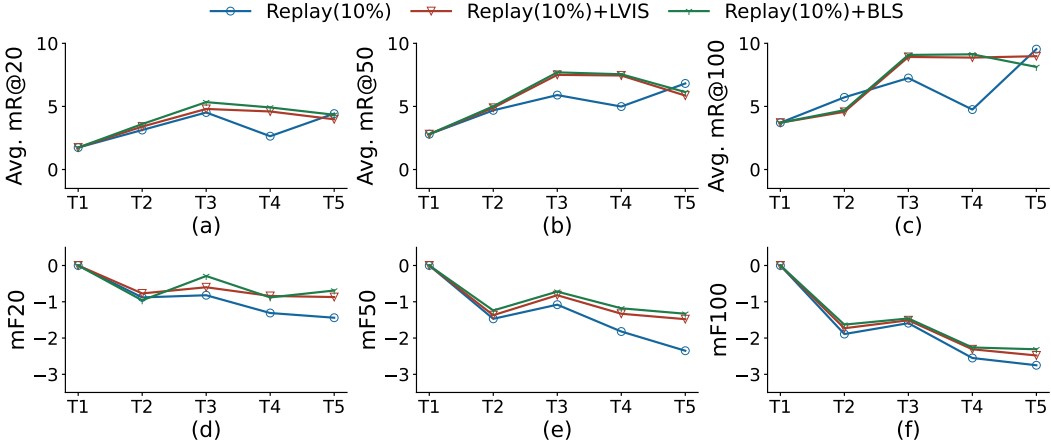

Figure S13: **Results in Avg. mR@K and mF@K over tasks in Learning Scenario 1 (S1) when sampling techniques on long-tailed distribution are applied.**. See **Sec. 3.2** for the introduction to techniques used for long-tailed distributions. See **Tab. 4** for design conventions.

### A.5.3   RESULT ANALYSIS ON BWT@K AND FWT@K

We provide the results of BWT@K and FWT@K in **Fig. S14** for Learning Scenario 1. Consistent with the results of Avg.R@20 and F@20, we observe that among all methods, the naive method has the lowest FWT@20 and BWT@20 implying catastrophic forgetting. Once again, EWC outperforms the naive baseline by having higher FWT@20 and BWT@20 though the difference is small as it fails in longer task sequences. Although all the baselines have negative BWT@20 indicating forgetting, Replay(10%) yields a higher BWT@20 than PackNet, EWC and naive baseline pointing to a reduced level of forgetting. Replay(100%) exhibits the greatest BWT@20, suggesting the significance of revisiting previous samples to prevent forgetting and facilitate backward knowledge transfer. Similar to BWT@20, most of the baselines have negative FWT@20 implying that knowledge acquired from prior tasks slightly interferes with the learning process of the current task. In contrast, Replay(10%) shows positive FWT@20 indicating that knowledge acquired from prior tasks enhances the learning process of the current task. Surprisingly, while Replay(100%) excels in Avg.R@20, F@20, and BWT@20, Replay(10%) achieves a superior FWT@20. This suggests that Replay(100%) might not grasp the current task as effectively as Replay(10%), as it preserves the knowledge from previous tasks by replaying more old samples and forgetting less at the sacrifice of slow learning about the current task. Similar trends are observed and the same analysis can be applied in Learning Scenario 2 (**Fig. S15**).

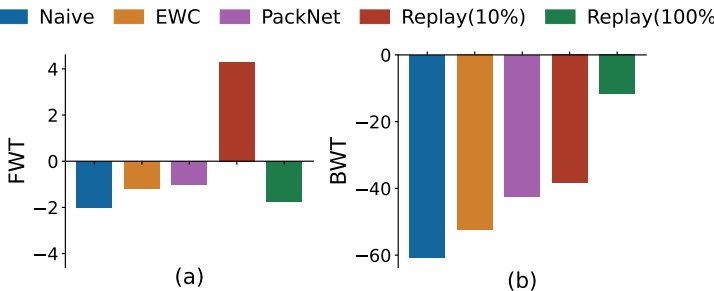

Figure S14: **Results of FWT (a) and BWT (b) over baselines in Learning Scenario 1 based on the SGTR backbone**. See **Sec. 3.2** for introduction to continual learning baselines. See **Sec. 3.3** for explanations about evaluation metrics. The x-axis indicates the various baselines. The higher FWT@20 and BWT@20, the better.

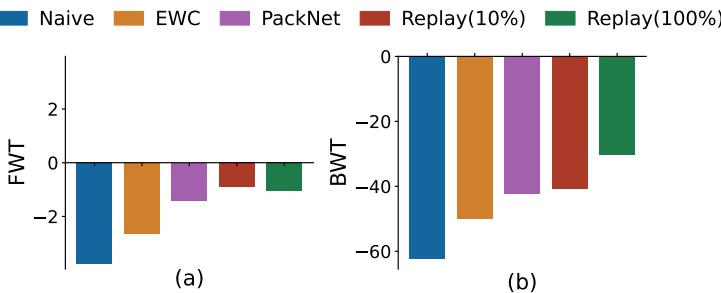

Figure S15: **Results of FWT (a) and BWT (b) over baselines in Learning Scenario 2 based on the SGTR backbone**. See **Sec. 3.2** for introduction to continual learning baselines. See **Sec. 3.3** for explanations about evaluation metrics. The x-axis indicates the various baselines. The higher FWT@20 and BWT@20, the better.

### A.5.4 RESULTS FOR DIFFERENT TASK SEQUENCES

Recent work (Singh et al., 2022) has highlighted the consistent and substantial curriculum effects in class-incremental learning and continual visual question-answering tasks. Inspired by these findings, we conducted experiments to assess the impact of the curricula in the context of CSEGG. To delve into its potential influence, we trained baselines with three distinct task sequences for Learning Scenario 1. Our results demonstrated that curriculum learning indeed shapes CSEGG performance within class-incremental settings. Notably, in **Fig. S16(d)**, a large difference in Avg.R@20 between Order 1 and Order 3 emerges for the Replay (100%) baseline. Similarly, **Fig. S16(a)** reveals a substantial Avg.R@20 disparity between Order 2 and Order 3 for the Naive baseline. This trend extends to F@20, as depicted in **Fig. S16(e)(f)(g)(h)**. These insights collectively affirm the significance of the curricula within CSEGG.

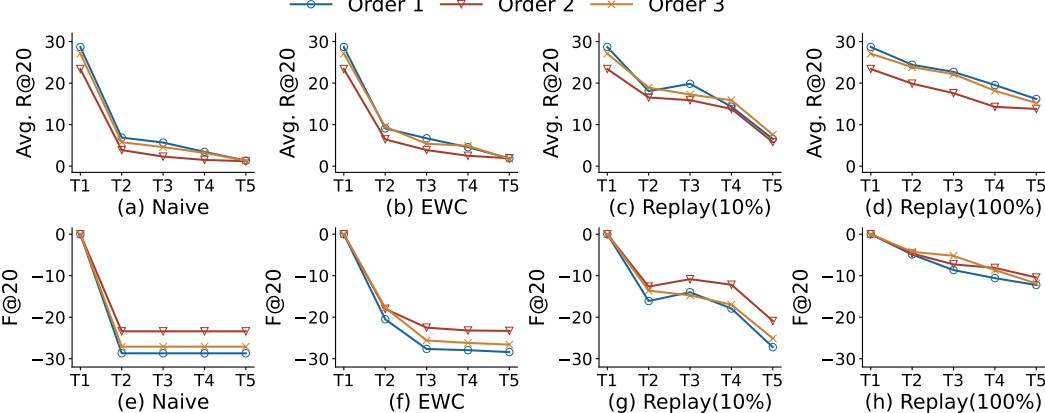

Figure S16: **Results of F@K=20, Avg. R@K=20 over tasks on CSEGG models with the SGTR backbone in Learning Scenario 1 with different permutations of task sequences.** (a),(e) denotes Avg.R@K and F@K for naive baseline. (b),(f) denotes Avg.R@K and F@K for EWC baseline. (c),(g) denotes Avg.R@K and F@K for Replay(10%) baseline. (d),(h) denotes Avg.R@K and F@K for Replay(100%) baseline. See **Sec. 3.2** for introduction to continual learning baselines. See **Sec. 3.3** for explanations about evaluation metrics. X-axis indicates the task numbers. The higher F, Avg.R@20, the better.

### A.5.5 MINIMAL FORGETTING IN DETR

To validate the impact of fine-tuning the DETR model in training Stage 2 of learning scenario S1 on relationship predicate predictions and to ensure minimal forgetting occurs in object detection (**Sec A.2**), we compare the mean Average Precision (mAP) for object detection on the entire test set of Visual Genome between the pre-trained DETR checkpoint from the paper (Li et al., 2022b) and the DETR models after fine-tuning on each task of S1.

As shown in **Fig. S17**, the results indicate a slight decrease of 0.4 in mAP from the pre-trained checkpoint to the DETR models over 5 tasks. This study provides evidence that fine-tuning DETR in S1 has negligible effects on forgetting. The forgetfulness observed in S1 can only be attributed to relationship incremental learning.

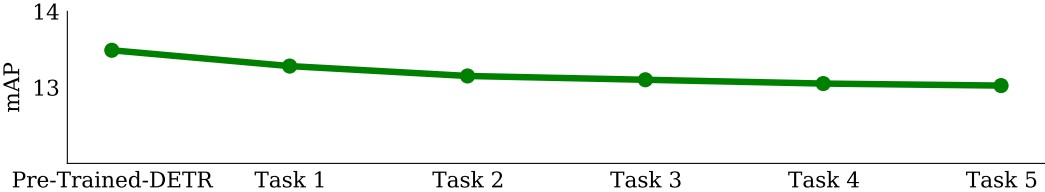

Figure S17: **mAP performance on the entire Visual Genome Test Set.** We used a pre-trained DETR checkpoint (Li et al., 2022b) as well as a naive baseline of SGTR continually trained on each task from S1, evaluated them on the entire Visual Genome's test set, and reported their mAP performances. We observed a minimal decrease in mAP across all tasks.

A.6    COMPARISION BETWEEN CNN-SGG BACKBONE AND THE SGTR MODEL

Based on the experimental results in **Fig. S18** for learning scenario 1, we observed similar relative performance variations across various CSEGG methods in **Sec. 5.1**. We also noticed a decrease in absolute performance in Avg.R and F for the CNN-SGG based backbone compared with the SGTR-based backbone. Similar trends and reasonings can be applied in Learning Scenario 2 (**Fig. S19**). However, in **Fig. S20** in Scenario 3, we noticed minimal performance differences in Gen $R_{bbox}$ and Gen R. This implies that the models with both backbones can generalize to detect unknown objects and recognize known relationships among unknown objects equally well.

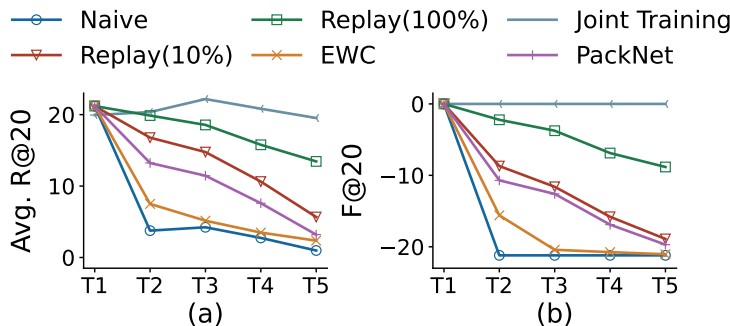

Figure S18: **Results of average recall and forgetting over tasks in Learning Scenario 1 (a and b) for CNN-SGG, when K-20.** See **Sec. 3.2** for introduction to continual learning baselines. See **Sec. 3.3** for explanations about evaluation metrics. X-axis indicates the task numbers. The higher Avg.R@20 and F@20, the better.

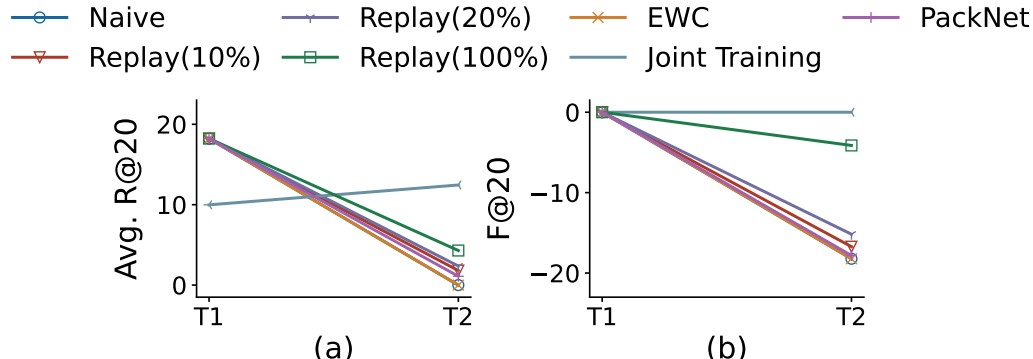

Figure S19: **Results of average recall and forgetting over tasks in Learning Scenario 2 (a and b) for CNN-SGG (K=20).** See **Sec. 3.2** for introduction to continual learning baselines. See **Sec. 3.3** for explanations about evaluation metrics. X-axis indicates the task numbers. The higher Avg.R@20 and F@50, the better.

**Fig. S20** provides the generalization results in detecting unknown objects and classifying known relationships among these objects in Learning Scenario 3 (S3) for CNN-SGG backbone. As elaborated in **Sec. 5.3**, an intriguing trend emerges: the CSEGG models demonstrate an increasing ability to identify unknown objects as the number of tasks rises. Furthermore, similar to our observations in **Sec. 5.3**, the CNN-SGG model exhibits proficiency in identifying known relationships between unfamiliar objects. However, this capability reaches a plateau as the task count escalates.

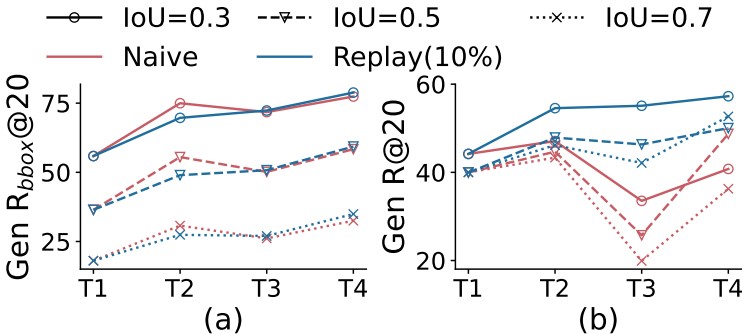

Figure S20: **Generalization results in Learning Scenario 3 for CNN-SGG (K=20).** See **Sec. 3.3** for evaluation metrics. The higher the values, the better. Line colors indicate continual learning baselines. Line types denote the IoU thresholds for determining correctly predicted bounding box locations.

## A.7 VISUALIZATION EXAMPLES FOR ALL LEARNING SCENARIOS FOR CONTINUAL SGTR-BASED MODELS

In this section, we present visualization examples from each learning scenario to showcase the performance of the three continual SGTR-based models, namely Replay10%, EWC, and Naive, in three learning scenarios.

### A.7.1 LEARNING SCENARIO 1 (S1)

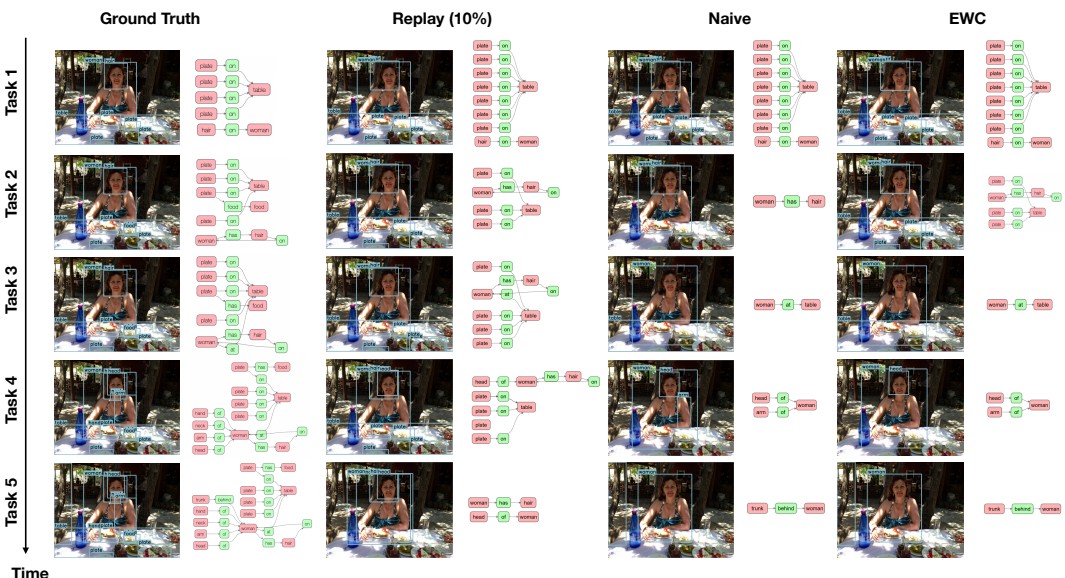

Figure S21: **Visualization example for Learning Scenario 1 (S1).** The leftmost column in the figure displays the ground truth bounding boxes and scene graphs for each task in Learning Scenario 1 (S1). The remaining columns, from left to right, represent the bounding boxes and scene graphs generated by each baseline model (Replay 10%, Naive, and EWC). In all the scene graphs, red boxes indicate objects, while green boxes represent relationships. The direction of the arrows between the red (object) and green (relationship) boxes indicates the subject and object ordering in the triplet. For example, in the scene graph predicted by the EWC model after Task 5, the triplet is "trunk behind women", as the arrow goes from "trunk" to "behind" to "women". The time arrow on the left side of the figure demonstrates that the model is exposed to new data over time, with new relationships incrementally added, as described in **Sec. 3.1**.

From **Fig.S21** we observe that, in Task 1, the ground truth scene graph contains triplets of "on" relationship: "plate on table" and "hair on women". After training on task 1, all three models (Replay 10%, EWC, Naive) can accurately predict these triplets of "on" relationship.

In Task 2, triplets of "has" relationship are introduced: "plate has food" and "women has hair". After training on task 2 data, the Replay 10% model successfully remembers the triplets of "on" relationship ("plate on table", "hair on women") from Task 1 and predicts "women has hair". The Naive model forgets the triplets of "on" relationship and only predicts "women has hair". The EWC model remembers the triplets of "on" relationships and predicts "women has hair". None of the models predict "plate has food".

In Task 3, triplet of "at" relationship is introduced: "women at table". After training on task 3 data, the Replay 10% model remembers the previous triplets ("plate on table", "women has hair", "hair on women") and predicts "women at table". The Naive model forgets the previous triplets and only predicts "women at table". In contrast to its previous performance, the EWC model forgets the previous triplets and only predicts "women at table".

In Task 4, triplets related to "of" relationship are introduced: "hand of women", "neck of women", "arm of women", and "head of women". After training on task 4 data, the Replay 10% model remembers the triplets related to "on" and "has" relationships ("plate on table", "women has hair",

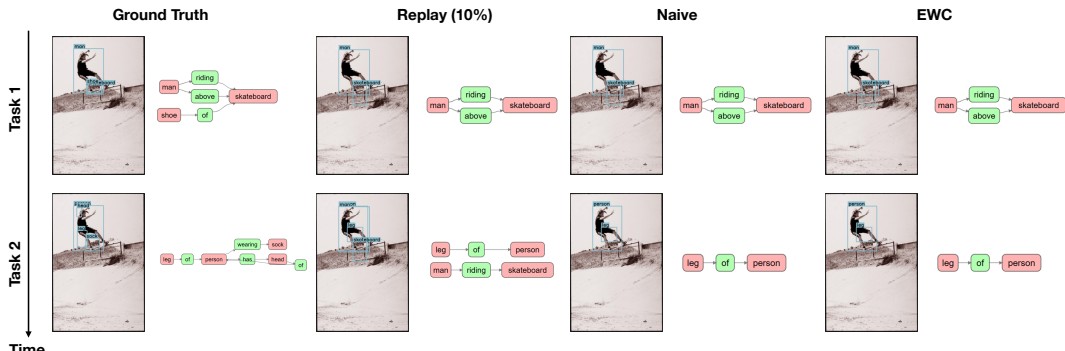

Figure S22: **Visualization example for Learning Scenario 2 (S2).** The leftmost column shows the ground truth bounding boxes and scene graphs in each task of Learning Scenario 2 (S2). The remaining columns, from left to right, represent the bounding boxes and scene graphs generated by each baseline model (Replay 10%, Naive, and EWC). In all the scene graphs, red boxes indicate objects, while green boxes represent relationships. As explained in **Fig. S21** caption, the direction of the arrows between the red (object) and green (relationship) boxes indicates the subject and object ordering in the triplet. The time arrow on the left side of the figure demonstrates that the model is exposed to new data over time, with new objects and relationships incrementally added, as described in **Sec. 3.1**.

"hair on women") from previous tasks but forgets the "at" relationship triplet ("women at table"). It only predicts "head of women" from the triplets introduced in Task 4. The Naive and EWC models both forget the "at" relationship triplet from the previous task but predict "head of women" and "arm of women" from the triplets introduced in Task 4.

In Task 5, triplet belonging to the "behind" relationship is introduced: "trunk behind women". After training on task 5 data, the Replay 10% model forgets the triplets related to "on" relationship ("plate on table", "hair on women") and only remembers the triplets related to "has" and "of" relationships ("women has hair", "head of women") learned from the previous task. It is not able to predict "trunk behind women". The Naive model, similar to its performance after previous tasks, fails to remember any triplets previously learned and only predicts "trunk behind women". The EWC model also fails to remember any triplets from the previous task and only predicts "trunk behind women".

### A.7.2    LEARNING SCENARIO 2 (S2)

From **Fig.S22** we observe that, in Task 1, the ground truth scene graph contains triplets: "man riding skateboard", "man above skateboard", and "shoe of skateboard". After training on task 1 data, all three models (Replay 10%, Naive, EWC) predict "man riding skateboard" and "man above skateboard". None of the models predict "shoe of skateboard".

In Task 2, new triplets introduced are: "leg of person", "person wearing sock", "person has head", and "head of person". After training on task 2 data, the Replay 10% model only remembers "man riding skateboard" from the previous task, forgetting "man above skateboard". Moreover, Replay 10% model can only predict "leg of person" from the triplets introduced in task 2. The Naive model forgets all the triplets from task 1 ( "man riding skateboard", "man above skateboard") and only predicts "leg of person" from the triplets introduced in task 2. Similar to Naive model, EWC model forgets all the triplets from task 1 ( "man riding skateboard", "man above skateboard") and only predicts "leg of person" from the triplets introduced in task 2.

### A.7.3    LEARNING SCENARIO 3 (S3)

**Fig. S23** illustrates the performance of the Replay 10% and Naive models in locating unknown objects and recognizing the relationships between these objects and other nearby unknown objects. The ground truth in **Fig. S23** consists of three unknown objects: "mountain", "sheep", and "house", along with three relationships: "near", "behind", and "infront of" (between "mountain" and "house"), and a "near" relationship (between "sheep" and "house").

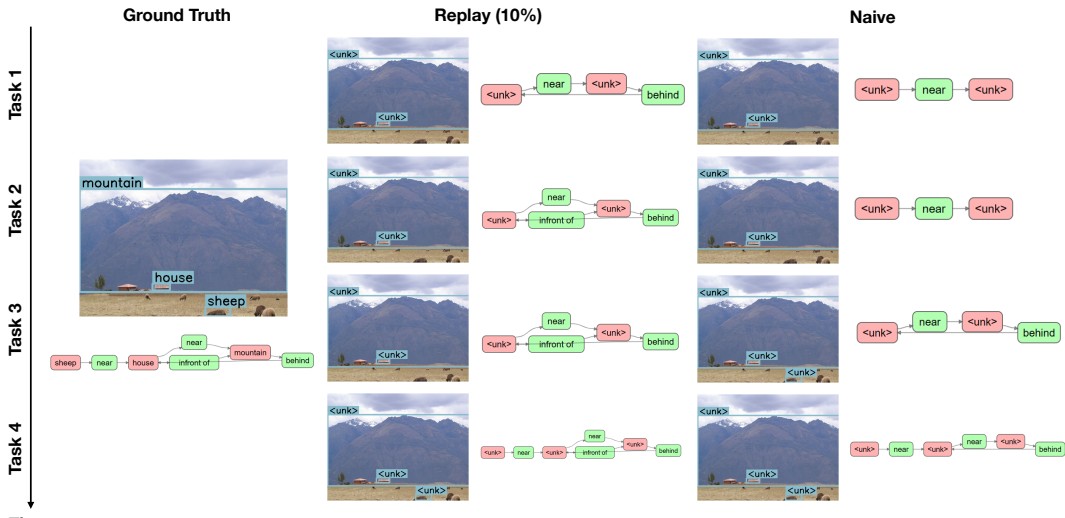

Figure S23: **Visualization example for Learning Scenario 3 (S3).** The leftmost column shows the standalone ground truth bounding boxes and scene graphs in the generalization test set regardless of which task it is in Learning Scenario 3 (S3). The remaining columns, from left to right, represent the bounding boxes and scene graphs generated by each baseline model (Naive, Replay 10%). Similar to **Fig. S21**, and **S22**, the red boxes in all scene graphs indicate objects , while green boxes represent relationships. As explained in **Fig. S21** caption, the direction of the arrows between the red (object) and green (relationship) boxes indicates the subject and object ordering in the triplet. The time arrow on the left side of the figure demonstrates that the model is exposed to new objects over time as described in **Sec. 3.1**. For easy referral of object instances in the predicted scene graphs, we numbered the unknown bounding boxes in this figure, where the numbers are not actually present in the model predictions. Concretely, unk_1 refers to "mountain"; unk_2 is "house"; and unk_3 is "sheep".

After training on Task 1 data, the Naive model can accurately locate the objects "mountain" and "house". However, no new object, such as "sheep", is located by the Naive model after training on Task 2 data. After training on Task 3 data, the Naive model is also able to locate the object "sheep" in addition to "mountain" and "house". Even after training on Task 4 data, the Naive model continues to locate all three objects: "mountain", "house", and "sheep". In contrast, the Replay 10% model, after training on Task 1 data, can only locate "mountain" and "house". This remains the same even after training on Task 2 and Task 3 data, where the Replay 10% model can still only locate the objects "mountain" and "house". However, after training on Task 4 data, the Replay 10% model is able to locate all the objects: "mountain", "house", and "sheep".

Regarding relationship generalization on unknown objects, the Naive model, after training on Task 1, can only predict the "near" relationship between the located objects "mountain" and "house" out of the three possible relationships. This performance remains the same even after Task 2. However, after training on Task 3, the Naive model can predict the "behind" relationship in addition to the "near" relationship between the located objects "mountain" and "house". After Task 4, the Naive model can predict the "behind" and "near" relationships between the located objects "mountain" and "house", as well as the "near" relationship between the located objects "sheep" and "house". In contrast, the Replay 10% model, after training on Task 1, can predict the "near" and "behind" relationships between the located objects "mountain" and "house". After Task 2, it can also predict the "infront of" relationship between the located objects "mountain" and "house" along with "near" and "behind" relationships. Even after Task 3, the Replay 10% model is still able to predict the "near", "behind", and "infront of" relationships between the located objects "mountain" and "house". After Task 4, as it can now locate the object "sheep", the Replay 10% model can also predict the "near" relationship between the objects "house" and "sheep", in addition to the existing relationships between "mountain" and "house".

