# OpenReview forum: "Adaptive Visual Scene Understanding: Incremental Scene Graph Generation"
_ICLR.cc/2024/Conference — ICLR 2024 Conference Withdrawn Submission_

### Official Review · Reviewer_SDSd · 2023-10-28

**Soundness:** 3 good
**Presentation:** 2 fair
**Contribution:** 3 good
**Rating:** 6
**Confidence:** 4

**Summary:**

The paper investigates different approaches towards continual learning of image scene graphs, where the continual learning of object vocabulary, predicate vocabulary and the combination can increase over time. The authors have experimented with several baselines, proposed several metrics ranging from Recall to mRecall; forgetfulness to generalizability with increasing task numbers.

**Strengths:**

The problem of image scene graph generation is inherently long-tailed. On top of that, the continuous learning on objects and predicates and their combination makes the problem even more challenging. Their experiments are methodical. The overall idea of the continual learning of them are based on the observation that these are long-tailed distributions and the task definition, dataset creation are all motivated by standard long-tailed learning paradigms which long-tailed class-incremental learning.

**Weaknesses:**

The writing is convoluted in some places. Specially the scene graph generation backbone 3.2 wasn't clear after the first read. The references to CNN-SGG and SGTR aren't separated, and caused a bit of confusion.

**Questions:**

As evident by several studies, the mean recall usually improves at the cost of recall in SGG literature. What is the impact of the long-tailed incremental learning in the forgetfulness of Recall? A recent paper proposed a one-stage method [1] which provided a good balance between Recall and mean Recall. Is it possible to utilize similar backbone so we know the effect of incremental learning on both Recall and mean Recall in a balanced way. Table 2 in [1] shows that mean recall of [1] is more than twice than that of training baseline of Fig S.8 in the current paper.


[1] Desai et al, "Single-Stage Visual Relationship Learning using Conditional Queries", NeurIPS 2022.

---

> ### Author Response · Authors · 2023-11-22
> **Official Comment by Authors**
>
> **SDSd.1 Weaknesses: The writing is convoluted in some places. Specially the scene graph generation backbone 3.2 wasn't clear after the first read. The references to CNN-SGG and SGTR aren't separated, and caused a bit of confusion.**
>
> We acknowledge the reviewer's feedback regarding the convoluted explanation of the scene graph generation backbones. To provide a clearer understanding, we offer a more concise and separate explanation for each backbone.
>
> For SGTR, the approach uniquely formulates the task as a bipartite graph construction problem. Starting with a scene image (Ii), SGTR utilizes a 2D-CNN and transformer-based encoder to extract image features. These features are then incorporated into a transformer-based decoder, predicting object and subject nodes (Oi). Predicate nodes (Ri) are formed based on both image features and object node features, and a bipartite graph (Gi) is constructed to represent the scene collectively. The correspondence between object nodes (oi) and predicate nodes (rk) is established using the Hungarian matching algorithm (Kuhn, 1955). Experimental results are based on the average over three runs, and the implementation leverages public source codes from (Li et al., 2022b) and (Wang et al., 2021b) with default hyperparameters. Refer to Section A.2.1 for detailed training and implementation specifics.
>
> As for CNN-SGG, it employs Faster-RCNN (Girshick, 2015) to generate object proposals from a scene image (Ii). The model extracts visual features for nodes and edges from these proposals, and through message passing, both edge and node GRUs output a structured scene graph. The training occurs in two stages within a supervised framework. In the initial stage, only object detection losses from Faster-RCNN (Ren et al., 2015) are applied to Oi, incorporating cross-entropy loss for object class and L1 loss for bounding box offsets. In the subsequent stage, the visual feature extractor (VGG-16, pre-trained on ImageNet) and GRU layers are trained to predict final object classes, bounding boxes, and relationship predicates using cross-entropy loss and L1 loss.
>
> We have also clearly separated the explanation of each backbone in the revised main manuscript in Sec. 3.2. We have clearly referenced the training and implementation details for both backbones in the revised main manuscript as well in Sec. 3.2.

---

> > ### Author Response · Authors · 2023-11-22
> > **Official Comment by Authors**
> >
> > **SDSd.1 Questions: As evident by several studies, the mean recall usually improves at the cost of recall in SGG literature. What is the impact of the long-tailed incremental learning in the forgetfulness of Recall? A recent paper proposed a one-stage method [1] which provided a good balance between Recall and mean Recall. Is it possible to utilize similar backbone so we know the effect of incremental learning on both Recall and mean Recall in a balanced way. Table 2 in [1] shows that mean recall of [1] is more than twice than that of training baseline of Fig S.8 in the current paper.**
> >
> > **[1] Desai et al, "Single-Stage Visual Relationship Learning using Conditional Queries", NeurIPS 2022.**
> >
> >
> > We appreciate the thoughtful question raised by the reviewer regarding the impact of long-tailed incremental learning on the forgetfulness of Recall. Analyzing this impact is indeed a complex task, and we acknowledge the challenges posed by the inherent long-tailed distribution in Visual Genome (VG). Unfortunately, achieving a direct comparison with balanced (non-long-tailed) incremental learning in VG is not feasible due to the dataset's intrinsic distribution.
> >
> > However, in our exploration, we employed long-tailed sampling techniques, specifically BLS and LVIS, to mitigate the long-tailed distribution to some extent. This allowed us to observe certain trends. Notably, the application of these techniques resulted in a decrease in forgetting in Recall (R) during incremental learning, as evidenced by the data presented in Tab. 4 even though forgetting in mR also decreased as evidenced by Figure S11 (Old S8).
> >
> > A compelling observation emerged from the application of long-tailed sampling techniques in the exemplar (Figure S7 (Old S4) (b) and (d), and Figure S4 (Old S1) (c)). This intervention increased the representation of tail classes in the exemplar, leading to improved performance on both tail and head classes in the test set. Consequently, the recall gain from enhanced tail class predictions contributed to an overall increase in recall value across tasks, reducing the forgetting in recall. This also explains the decreased forgetting in the mR curves in Figure S11 (Old S8), as the performance on the tail classes improves, therefore the overall mR improves and Forgetting therefore decreases.
> >
> > In interpreting these observations, we hypothesize that recall will continue to increase until the frequency of tail classes approaches that of head classes. Beyond this point, we anticipate a decrease in recall due to a drop in the performance of head classes under more balanced evaluation conditions. Despite this anticipated drop, we expect the recall value to remain higher than observed in our results, indicating lesser forgetting if the data were balanced. In the case of mR, we hypothesize the same that if the data were balanced the forgetting in the mR case would be lesser in comparison to the current trends in Figure S11 (Old S8). This is self explanatory, as the data is more balanced, the mR would be higher and hence the forgetting will be lower.
> >
> > In summary, our observations suggest that the long-tailed distribution tends to increase forgetting in both R and mR, and achieving a more balanced dataset would likely result in improved R and mR performance and reduced forgetting.
> >
> > We greatly appreciate the reviewer's insightful suggestion and the introduction to the work presented in [1]. We find the idea of implementing such backbones in the CSEGG setting intriguing, as it provides an opportunity to study the effects of long-tailed distribution on recall and mean recall in an incremental learning scenario in a more balanced manner. This suggestion aligns well with our goals, and we plan to incorporate the backbone introduced in [1] into our future work on CSEGG. This will allow us to conduct a more comprehensive analysis and gain deeper insights into the impact of long-tailed distribution on different performance metrics. We thank the reviewer for this valuable input, and we are committed to exploring this avenue in our ongoing research.
> >
> > [1] Desai et al, "Single-Stage Visual Relationship Learning using Conditional Queries", NeurIPS 2022.

---

### Official Review · Reviewer_FdjE · 2023-10-30

**Soundness:** 2 fair
**Presentation:** 3 good
**Contribution:** 2 fair
**Rating:** 3
**Confidence:** 4

**Summary:**

This paper investigates the continual learning problem in SGG, and conducts experiments to show the performance of existing methods in the proposed continual learning scenarios.

**Strengths:**

1. The proposed scenario is realistic and is worth to investigate.
2. The paper is easy to follow.

**Weaknesses:**

1. Clarity on Relationship with SGG Tasks: The paper lacks clarity in establishing the direct correlation between its observations and the Scene Graph Generation (SGG) tasks. While the identified issues like catastrophic forgetting, the efficacy of replay methods, and addressing long-tail problems are extensively explored in existing research, the unique challenges in integrating continual learning, long-tail problems, and SGG remain unclear. The paper falls short in delineating the specific challenges arising from the amalgamation of these factors.

2. Inadequate Support for Conclusions: The paper argues that replay-based methods underperform on S3 due to models focusing on detecting more in-domain object boxes. However, this conclusion lacks direct substantiation. The mixed nature of the test datasets across all tasks in S3 complicates such straightforward assertions. Further information and experimental evidence are necessary to strengthen and clarify this particular assertion.

3. Limited Contribution to Method Design: The primary contribution of this paper lies in evaluating continual learning algorithms within the proposed scenarios. Nevertheless, this contribution, while valuable, may not meet the rigorous criteria for publication in esteemed venues such as ICLR. This is primarily because it lacks the introduction of novel methods or groundbreaking observations that can serve as a source of inspiration and guidance for future method development.

**Questions:**

1. Could the authors delve further into elucidating the unique challenges that arise in SGG tasks due to continual learning scenarios? Clarifying these challenges could significantly strengthen the paper's contribution.

2. It would be beneficial if the paper explored how the insights garnered from this paper could inspire or guide the design of enhanced methodologies for continual SGG scenarios.

---

> ### Author Response · Authors · 2023-11-22
> **Official Comment by Authors**
>
> **FdjE.1 Weaknesses: Clarity on Relationship with SGG Tasks: The paper lacks clarity in establishing the direct correlation between its observations and the Scene Graph Generation (SGG) tasks. While the identified issues like catastrophic forgetting, the efficacy of replay methods, and addressing long-tail problems are extensively explored in existing research, the unique challenges in integrating continual learning, long-tail problems, and SGG remain unclear. The paper falls short in delineating the specific challenges arising from the amalgamation of these factors.**
>
> **FdjE.1 Questions: Could the authors delve further into elucidating the unique challenges that arise in SGG tasks due to continual learning scenarios? Clarifying these challenges could significantly strengthen the paper's contribution.**
>
> We appreciate the reviewer's feedback and would like to provide a detailed response to address the concerns. Challenges faced by us are as follows :-
>
> * Creating diverse tasks for CSEGG poses a non-trivial challenge, primarily due to the intricate relationships between objects and the potential entanglement of tasks. An essential aspect of task creation involves meticulous data analysis to address this complexity. For instance, dividing objects like "Man" and "Horse" into separate tasks can diminish the frequency of the relevant relationship "riding" since a new task may lack either "Man" or "Horse." Blindly partitioning the dataset based on random assignment of objects or relations to tasks risks eliminating numerous pertinent triplets. Careful analysis was indispensable in crafting each learning scenario to ensure the maximum inclusion of relevant training examples in each task. Consider, for instance, a task with objects like "Horse," "Car," and relationships like "on." Such a scenario fails to yield contextually relevant triplets. Therefore, meticulous data analysis was crucial during the creation of each learning scenario. In designing Learning Scenario 1, preserving the long-tailed nature of the dataset across all tasks was a priority. This was imperative to prevent an imbalance where one task would be overloaded with long-tailed classes, leaving the remaining tasks with a shortage of training examples. In Learning Scenario 2, the division of objects and relations was carefully orchestrated to maximize the number of images for both tasks while ensuring the inclusion of contextually relevant triplets. In the case of CSEGG, Learning Scenario 3 involved a detailed analysis to determine the optimal number of objects in each task. This aimed at maximizing the number of training examples and ensuring an equitable distribution of common relations across all tasks. The intricate analysis undertaken to design each task within the learning scenarios is a pivotal challenge specific to CSEGG. Along with this, we overcame implementation challenges as we are the first to implement various SGG backbones in the continual learning setting.
> * A significant hurdle we encountered in our study was the absence of continual learning methods specifically tailored to minimize forgetting and optimize the performance of CSEGG models. In response to this challenge, we introduced a novel generative replay method termed "Replays via Analysis by Synthesis" (RAS) detailed in Sec. 4. This method harnesses the capabilities of generative models to artificially produce exemplars representing the classes present in each task. Subsequently, these exemplars are employed to train the model in subsequent tasks, effectively mitigating catastrophic forgetting. The application of RAS is particularly pertinent to SGG, as evident from our observations with CSEGG baselines (which is consistent with continual learning literature covering tasks like detection and classification), where replay methods exhibited superior performance in addressing catastrophic forgetting. However, practical considerations, such as limited storage and privacy concerns in real-world scenarios, posed challenges for traditional replay methods. To address these limitations, we adopted a generative replay approach that generates images solely from the textual descriptions of triplets in each task. This not only enhances user privacy but also allows for the generation of a flexible number of images, eliminating storage constraints as these images are used once and not stored. A unique challenge specific to CSEGG arose during this process. In contrast to image classification scenarios where generated images typically require no annotations, the case of SGG demands annotations for object locations, object labels, and relation labels. To surmount this challenge, we leveraged each task's trained model to generate annotations for the synthetically created images. This combined notation, along with the generated image, formed the basis for training the SGG model, effectively addressing the issue of catastrophic forgetting in the CSEGG context.

---

> > ### Author Response · Authors · 2023-11-22
> > **Official Comment by Authors**
> >
> > **FdjE.2 Weaknesses: Inadequate Support for Conclusions: The paper argues that replay-based methods underperform on S3 due to models focusing on detecting more in-domain object boxes. However, this conclusion lacks direct substantiation. The mixed nature of the test datasets across all tasks in S3 complicates such straightforward assertions. Further information and experimental evidence are necessary to strengthen and clarify this particular assertion.**
> >
> > We appreciate the reviewer's concern regarding the assertion that replay-based methods underperform on S3 due to models focusing on detecting more in-domain object boxes. We understand the need for clarification, and we would like to address the points raised by the reviewer.
> >
> > Firstly, it's important to clarify that there is no mixed test dataset in the case of S3. As stated in Section 3.1 of the manuscript, there is a standalone test set for S3 that is consistent across all four tasks. This test set comprises 30 entirely unseen objects (not present in the training data) and a common set of relationships found in the training data. This design ensures uniformity in the test datasets across tasks in S3.
> >
> > With this clarification, we can assert that the claim regarding the underperformance of replay-based methods on S3, specifically on R_bbox, is supported by the fact that the models are tested on a single, common dataset of unknown objects. This ensures that any observed differences in performance can be attributed to the models' ability to handle new and unseen object classes, providing a more straightforward basis for the conclusion.

---

> > > ### Author Response · Authors · 2023-11-22
> > > **Official Comment by Authors**
> > >
> > > **FdjE.3 Weaknesses: Limited Contribution to Method Design: The primary contribution of this paper lies in evaluating continual learning algorithms within the proposed scenarios. Nevertheless, this contribution, while valuable, may not meet the rigorous criteria for publication in esteemed venues such as ICLR. This is primarily because it lacks the introduction of novel methods or groundbreaking observations that can serve as a source of inspiration and guidance for future method development.**
> > >
> > > **FdjE.2 Questions: It would be beneficial if the paper explored how the insights garnered from this paper could inspire or guide the design of enhanced methodologies for continual SGG scenarios.**
> > >
> > > Certainly, our evaluation process not only provided insights into the performance of existing continual learning methods but also inspired the creation of an innovative solution called "Replays via Analysis by Synthesis" (RAS)  to tackle the significant challenge of catastrophic forgetting in Continual Scene Graph Generation (CSEGG). This method was conceived based on the findings presented in Sec 5.1 and Tab 2 of our benchmarking results. The notable efficacy of rehearsal-based methods, particularly Replay 10% and Replay 20%, highlighted the importance of revisiting and reinforcing past task data points for the ongoing learning of new objects and relationships in CSEGG. However, recognizing the limitations associated with fixed-size exemplars in rehearsal-based methods, we explored alternative approaches.
> > >
> > > Given the constraints of fixed-size exemplars discussed in the continual learning literature, RAS was carefully designed as a generative modeling-based approach. To overcome the fixed-size challenge, the method opts to store unique triplets in a text file for each task instead of multiple images and annotations, prioritizing memory efficiency. Leveraging advanced generative image models such as the Stable Diffusion model, this approach generates images from these stored triplets, serving as exemplar examples for continual learning. The subsequent step involves utilizing current task’s trained SGG model to generate notations for these generated images, completing the exemplar, which is then used to train with new data for subsequent tasks. This is explained in detail in Sec. 4 of the manuscript.
> > >
> > > To refine the specifics of our method, ablation studies were conducted, as detailed in Tab. 3. This study aimed to make major design choices for our method, including experiments to select the diversity present in the image along with identifying the optimal strategy for generating images for the exemplar. The key findings are listed below :-
> > > * As illustrated in Tab. 3 by comparing (A1 and A2), then (A3 and A4) and finally can also be seen in (A5 and A6), images generated from multiple triplets outperformed those from a single triplet, a trend observed in both Visual Genome (VG) images and the generated images.
> > > * Another experiment assessed the quality of notations produced by each task’s model by generating notations for VG images and using them in the exemplar alongside VG images. As shown in Tab. 3 by comparing A2 and A4, applying the same approach to VG images resulted in a minimal drop in Avg.R (16.8%) and F (7%) for the last task of Scenario 1, confirming the quality of the generated notations.
> > > * Additional experimentation compared the performance of generated images with VG images, and as demonstrated in Tab. 3 by comparing A4 and A6, the generated images exhibited decent quality. The comparative analysis in Tab. 3 revealed that when using generated images instead of ground truth images, the drop in Avg.R and F was only 7.32% and 6.02%.
> > >
> > > While a detailed analysis of RAS results can be found in Sec. 5.4 of the main manuscript. The development of RAS stands as a testament to the transformative impact of insights derived from our benchmarking process. By addressing memory-based constraints inherent in conventional Replay methods through the integration of generative modeling and emphasizing the importance of rehearsal for continual learning of objects and relationships, this method illustrates how our benchmarks can drive the evolution of new and effective algorithms tailored to the challenges of the CSEGG setting.

---

### Official Review · Reviewer_pXA5 · 2023-10-31

**Soundness:** 3 good
**Presentation:** 2 fair
**Contribution:** 3 good
**Rating:** 6
**Confidence:** 4

**Summary:**

This work proposes comprehensive studies on several new settings of the scene graph generation task, in which the relationships, scene, and object incremental scenarios are considered. It conducts experiments that combine continuous learning with current two-stage and transformer-based SGG methods and analyzes their performance.

**Strengths:**

I think continual scene graph generation is quite practical.

The authors provide some introduction and analyses about the learning scenarios, evaluation methods and metrics, and results using current SGG algorithms combined with continuous learning. These analyses are quite essential.

**Weaknesses:**

It seems the organization and writing are quite disordered and difficult to follow. For example, the authors claim scenario 1 has 5 tasks. However, detailed definitions of these tasks are missing. Similar problems exist for the other two scenarios.

The first contribution seems to over-claim. It seems the images, object classes and relationships inherit from the visual genome dataset. I do not know what is new.

**Questions:**

see weakness

---

> ### Author Response · Authors · 2023-11-22
>
> **pXA5.1 Weaknesses: It seems the organization and writing are quite disordered and difficult to follow. For example, the authors claim scenario 1 has 5 tasks. However, detailed definitions of these tasks are missing. Similar problems exist for the other two scenarios.**
>
> We apologize to the reviewer for any confusion resulting from the organization and writing of the manuscript. The meaning of each learning scenario is now clarified in Sec. 3.1 of the main manuscript. Detailed definitions of all tasks for each learning scenario, along with data statistics, are provided in Sec. A.1 and Sec. A.3 and visualized in Figures S4, S5, S6. We acknowledge the oversight of not referencing these details in Sec. 3.1 and have rectified it in the revised manuscript by adding references to the detailed definition section for each learning scenario in Sec. 3.1.
>
>
> **pXA5.2 Weaknesses: The first contribution seems to over-claim. It seems the images, object classes and relationships inherit from the visual genome dataset. I do not know what is new.**
>
>
> We acknowledge the reviewer's concern regarding the first contribution and appreciate the opportunity to clarify. Our primary contribution, as outlined in "Main Contribution 1" in the main manuscript, is not centered around providing a new dataset with images and notations for Scene Graph Generation (SGG). Our focus uniquely centers on the intricacies and dynamics of continual learning within the domain of Scene Graph Generation (SGG). By introducing innovative splits for Visual Genome (VG), we have tailored the dataset to better support the demands of continual learning. Consequently, our efforts extend the capabilities of VG, resulting in the creation of the novel dataset, CSEGG. This dataset stands as the pioneering resource designed specifically for examining the nuances of continual learning within the realm of SGG.
>
> The challenge arises from the inherent long-tail nature of the VG dataset, making it non-trivial to divide. Our main contribution lies in carefully designing and creating the splits for each continual learning scenario, addressing the long-tailed distribution within VG. This involves selecting combinations of triplets that enable meaningful training samples for each task, ensuring the model can effectively learn from the dataset. For example, in creating Learning Scenario S3, detailed data analysis was conducted to identify splits that maximize common relations across tasks while also providing testing samples with entirely new objects and the same relations.
>
> Along with specific data division, the design of the individual scenarios is quite unique and specific to study a fixed purpose. Learning Scenario 1 (S1) simulates a real-world scenario where an agent learns new relations over time, focusing on the continual learning impact on SGG when introducing new relations. This is essential for a model to identify new relations between known objects without relearning them, as detailed in Sec. A.1.1. Learning Scenario 2 (S2) replicates scenarios in which new objects and relationships are gradually introduced over time. This study is crucial for real-world applications, ensuring a model can perform a specific task with new objects and relationships, as discussed in Sec. A.1.2. Learning Scenario 3 (S3) evaluates the model's generalization capability, testing its ability to apply existing knowledge to unseen data. This is vital for practical applications, such as a robot using existing spatial relation knowledge to navigate unfamiliar objects, detailed in Sec. A.1.3.
>
> In summary, our contribution is not the images or notations themselves but rather the thoughtful design of the continual learning scenarios and the specific data division within VG to analyze CSEGG models under the challenging conditions posed by the dataset's long-tailed nature.

---

### Official Review · Reviewer_LwTF · 2023-11-02

**Soundness:** 3 good
**Presentation:** 2 fair
**Contribution:** 2 fair
**Rating:** 6
**Confidence:** 4

**Summary:**

This work divides up the Visual Genome dataset into three scene graph-based continual learning scenarios. It then evaluates some baseline approaches on the benchmarks: two scene generation backbone architectures (SGTR and CNN-SGG), three sampling strategies for learning (LVIS, BLS, and EFL), as well as five continual learning approaches (naive, EWC, PackNet, Replay, and joint training).

**Strengths:**

* Scene graphs present an interesting (novel to my knowledge) domain for continual learning.
* The learning scenarios make sense and are well motivated with comparison to (somewhat distant) real-world scenarios.
* The work pays attention to the distribution of attributes/data for each task per learning scenario (e.g., relationships within a task are long-tailed).
* The choice of baselines and evaluation metrics seems sound to me.
* I particularly liked Figure 4 showing an overview of SGTR and the continual learning baseline algorithms.
* The authors have made their code available.

**Weaknesses:**

* The authors missed connections to the meta-learning and curriculum design literature. From that lens, claims such as "CSEGG methods improve generalization abilities" seem a bit unsurprising.
* The dataset is somewhat small in scale. 150 objects and 50 relationships might not be enough to pose a sufficient continual learning channel. The benchmark also reuses images between tasks in a learning scenario, which is not ideal.
* Though there is some interpretation of the baselines, I struggled to see what the community should take away about them.
* I'm also uncertain how the benchmark might encourage future algorithm or model developments.
* The writing introduces a lot of terms (in bold text). Some of this could be done better, for instance:
  * Please enumerate the continual learning algorithms in Section 3.3.
  * Some terms that are introduced are never referenced again in the main text (e.g., Forward and Backward Transfer).
* It is confusing when "long-tailed" in mentioned in the context of dataset creation as well as learning (e.g. the title of Section 4.2 is ambiguous. In Section 3.2, perhaps it would be clearer to say "Techniques for sampling to deal with long-tailed data"?).

**Questions:**

* There are other datasets which come with scene graphs as you laid out on your Related Work section. Why not use those?
* The caption for Fig 1 suggests objects are nodes and relations are edges. But Fig 1a shows them as nodes of different color. Could you please ensure consistency? Figure 1b intends to show that new objects and relations emerge over time, but some of the uncolored objects (e.g. man, tree) have not appeared previously. So which objects/relations are colored seems arbitrary?
* Could you please walk us through the immediate implications of your work for the community, rather than distant possibilities?

---

> ### Author Response · Authors · 2023-11-22
> **Official Response by Authors**
>
> **LwTF.1 Weaknesses: The authors missed connections to the meta-learning and curriculum design literature. From that lens, claims such as "CSEGG methods improve generalization abilities" seem a bit unsurprising.**
>
> We highly appreciate your detailed feedback, and we would like to provide additional clarification on how our work distinguishes itself from traditional meta-learning and incorporates curriculum design elements.
>
> Our primary focus is on evaluating the generalization testing performance of CSEGG models, specifically their ability to perform on unseen data. As outlined in Sec. 3.1, In S3 our models are trained across four tasks, each introducing 30 new object classes and a consistent set of 35 relationships. Notably, the object classes in each task are unique and not repeated in any other task. The test set is a standalone set comprising 30 objects not present in the training data, with only the relationships being common across all tasks and the standalone test. This design allows us to assess the model's capability to identify known relationships between objects it has never encountered during training. Importantly, our approach does not involve task adaptation during training and our model undergoes no training on data from the same domain as the test data, highlighting its domain independence and setting it apart from traditional meta-learning methods
>
> Furthermore, we have incorporated the impact of curriculum learning in Learning Scenario 1 (S1). In Sec. A.5.4 of the manuscript, we discuss the effect of curriculum learning in S1, identifying that curriculum learning affects the performances of CSEGG models as seen in Fig. S16. It also aligns with the conclusions in [1], where curriculum learning was found to have a substantial effect on class-incremental learning. We hope these clarifications address your concerns and provide a more comprehensive understanding of our approach.
>
>
> [1] Parantak Singh, You Li, Ankur Sikarwar, Weixian Lei, Daniel Gao, Morgan Bruce Talbot, Ying Sun, Mike Zheng Shou, Gabriel Kreiman, and Mengmi Zhang. Learning to learn: How to continuously teach humans and machines. arXiv preprint arXiv:2211.15470, 2022
>
> **LwTF.2 Weaknesses: The dataset is somewhat small in scale. 150 objects and 50 relationships might not be enough to pose a sufficient continual learning channel. The benchmark also reuses images between tasks in a learning scenario, which is not ideal.**
>
> We respectfully disagree with the reviewer's comments on the scale of the dataset. The term "small" is relative and context-dependent. For instance, MS COCO [1] is considered a substantial dataset in computer vision, playing a crucial role in numerous advancements in the field. While it may be deemed "small" when compared to ImageNet [2], its size is still significant and allows for quality research and development.
>
> Additionally, our choice of the Visual Genome (VG) dataset [3] is intentional and aligned with the specific requirements of our research. VG is a dedicated scene graph generation dataset designed for tasks similar to SGG. Our aim is to explore the effects of continual learning on SGG, a novel approach that hasn't been extensively studied before. The size of the dataset is not necessarily a limitation; rather, it aligns with the scope of our research objectives. VG provides a sufficient basis for studying continual learning in SGG, allowing us to investigate the model's performance across different tasks.
>
> The reuse of images between tasks is a deliberate choice, as explained in Sec. A.1.5 of our manuscript. We draw parallels to human learning scenarios, where a parent guides a baby to recognize various objects in a bedroom. Despite encountering the same scenes, the focus is on continual learning, instructing the baby to detect and recognize individual objects sequentially. This intentional reuse of images serves as a valuable aspect of our experimental design, reflecting real-world learning scenarios.
>
> [1] Lin, Tsung-Yi, et al. "Microsoft coco: Common objects in context." Computer Vision–ECCV 2014: 13th European Conference, Zurich, Switzerland, September 6-12, 2014, Proceedings, Part V 13. Springer International Publishing, 2014.
> [2] Deng, Jia, et al. "Imagenet: A large-scale hierarchical image database." 2009 IEEE conference on computer vision and pattern recognition. Ieee, 2009.
> [3] Krishna, Ranjay, et al. "Visual genome: Connecting language and vision using crowdsourced dense image annotations." International journal of computer vision 123 (2017): 32-73.

---

> > ### Author Response · Authors · 2023-11-22
> > **Official Response by Authors**
> >
> > **LwTF.3 Weaknesses: Though there is some interpretation of the baselines, I struggled to see what the community should take away about them.**
> >
> > * Our paper's primary contribution lies in the design of the Continual Scene Graph Generation (CSEGG) dataset. Unlike contribution of images or notations, our emphasis is on the thoughtful creation of continual learning scenarios and specific data division within Visual Genome (VG). This tailored dataset enables the analysis of CSEGG models under the challenging conditions posed by the long-tailed nature of the dataset, serving as a valuable resource for further developments and research in continual learning and CSEGG.
> > * Another significant contribution involves evaluating existing continual learning algorithms, such as Replay, Elastic Weight Consolidation (EWC), and PackNet, on established Scene Graph Generation (SGG) architectures. While trends like Replay excelling in mitigating catastrophic forgetting and EWC struggling on longer task sequences were observed in computer vision tasks like Image Classification, we are the first to validate these trends in the context of CSEGG. This work provides insights into the performance of these algorithms in the unique challenges posed by CSEGG.
> > * Additionally, our experiments reveal that continual learning in SGG models enhances model generalizability, as discussed in Sec. 5.3. In Learning Scenario 2 (S2), benchmarking experiments illustrate that while continual learning algorithms perform well in individual incremental object and relationship detection, they face challenges when both are combined. This underscores the need for addressing catastrophic forgetting in scenarios where new objects and relationships are introduced incrementally.
> > * Furthermore, our observations in Sec. 5.2 indicates that existing long-tail sampling techniques are not as effective in CSEGG as they are in SGG. This highlights the necessity of developing long-tailed distribution handling techniques specifically designed for CSEGG.
> >
> > These unique contributions contribute to advancing the understanding and capabilities of continual learning in the challenging domain of CSEGG.

---

> > > ### Author Response · Authors · 2023-11-22
> > > **Official Response by Authors**
> > >
> > > **LwTF.4 Weaknesses: I'm also uncertain how the benchmark might encourage future algorithm or model developments.**
> > >
> > >
> > > Certainly, our evaluation process not only provided insights into the performance of existing continual learning methods but also inspired the creation of an innovative solution called "Replays via Analysis by Synthesis" (RAS)  to tackle the significant challenge of catastrophic forgetting in Continual Scene Graph Generation (CSEGG). This method was conceived based on the findings presented in Sec 5.1 and Tab 2 of our benchmarking results. The notable efficacy of rehearsal-based methods, particularly Replay 10% and Replay 20%, highlighted the importance of revisiting and reinforcing past task data points for the ongoing learning of new objects and relationships in CSEGG. However, recognizing the limitations associated with fixed-size exemplars in rehearsal-based methods, we explored alternative approaches.
> > >
> > > Given the constraints of fixed-size exemplars discussed in the continual learning literature, RAS was carefully designed as a generative modeling-based approach. To overcome the fixed-size challenge, the method opts to store unique triplets in a text file for each task instead of multiple images and annotations, prioritizing memory efficiency. Leveraging advanced generative image models such as the Stable Diffusion model, this approach generates images from these stored triplets, serving as exemplar examples for continual learning. The subsequent step involves utilizing current task’s trained SGG model to generate notations for these generated images, completing the exemplar, which is then used to train with new data for subsequent tasks. This is explained in detail in Sec. 4 of the manuscript.
> > >
> > > To refine the specifics of our method, ablation studies were conducted, as detailed in Tab. 3. This study aimed to make major design choices for our method, including experiments to select the diversity present in the image along with identifying the optimal strategy for generating images for the exemplar. The key findings are listed below :-
> > > * As illustrated in Tab. 3 by comparing (A1 and A2), then (A3 and A4) and finally can also be seen in (A5 and A6), images generated from multiple triplets outperformed those from a single triplet, a trend observed in both Visual Genome (VG) images and the generated images.
> > > * Another experiment assessed the quality of notations produced by each task’s model by generating notations for VG images and using them in the exemplar alongside VG images. As shown in Tab. 3 by comparing A2 and A4, applying the same approach to VG images resulted in a minimal drop in Avg.R (16.8%) and F (7%) for the last task of Scenario 1, confirming the quality of the generated notations.
> > > * Additional experimentation compared the performance of generated images with VG images, and as demonstrated in Tab. 3 by comparing A4 and A6, the generated images exhibited decent quality. The comparative analysis in Tab. 3 revealed that when using generated images instead of ground truth images, the drop in Avg.R and F was only 7.32% and 6.02%.
> > >
> > > While a detailed analysis of RAS results can be found in Sec. 5.4 of the main manuscript. The development of RAS stands as a testament to the transformative impact of insights derived from our benchmarking process. By addressing memory-based constraints inherent in conventional Replay methods through the integration of generative modeling and emphasizing the importance of rehearsal for continual learning of objects and relationships, this method illustrates how our benchmarks can drive the evolution of new and effective algorithms tailored to the challenges of the CSEGG setting.
> > >
> > > **LwTF.5 Weaknesses: The writing introduces a lot of terms (in bold text). Some of this could be done better, for instance: (a) Please enumerate the continual learning algorithms in Section 3.3. (b) Some terms that are introduced are never referenced again in the main text (e.g., Forward and Backward Transfer).**
> > >
> > > We acknowledge the reviewer's feedback regarding the extensive introduction of terms in bold text and the potential challenges this may pose for readers. We have corrected this in revised manuscript.
> > >
> > > Regarding terms like Forward Transfer (FWT) and Backward Transfer (BWT), we acknowledge that detailed definitions and analysis were not provided in the main text due to space limitations. However, we introduced these terms in Sec. 3.3 and referenced the appendix Sec. A.2.3 for detailed definitions of these evaluation metrics. Furthermore, we referenced appendix Sec. A.5.3 in the same section of the main manuscript, where a detailed analysis of the results obtained from FWT and BWT metrics is provided. Along with this, results of FWT and BWT are also provided in Tab. 2.
> > >
> > > In addressing these concerns, we are open to revising the manuscript.  We appreciate the reviewer's feedback and will take proactive steps to improve the presentation of our work.

---

> > > > ### Author Response · Authors · 2023-11-22
> > > > **Official Response by Authors**
> > > >
> > > > **LwTF.6 Weaknesses: It is confusing when "long-tailed" in mentioned in the context of dataset creation as well as learning (e.g. the title of Section 4.2 is ambiguous. In Section 3.2, perhaps it would be clearer to say "Techniques for sampling to deal with long-tailed data"?).**
> > > >
> > > > We recognize the potential confusion arising from the use of "long-tailed" in both the context of dataset creation and learning. To address this concern, we have revised the title of Section 5.2 (4.2 of the old manuscript) to read "Analyzing the Impact of Sampling Techniques on Long-Tailed Distribution in CSEGG." Additionally, we have updated the title in Section 3.2 to "Sampling Methods to Handle Long-Tailed Data" to provide clearer and more accurate descriptions of the content within each section. These changes aim to enhance the overall clarity and comprehension of the manuscript.
> > > >
> > > > **LwTF.1 Questions: There are other datasets which come with scene graphs as you laid out on your Related Work section. Why not use those?**
> > > >
> > > > We have chosen to utilize the Visual Genome (VG) dataset for several reasons, making it our preferred choice among other datasets with scene graphs. Visual Genome is widely recognized within the computer vision community as a prominent benchmark for evaluating Scene Graph Generation (SGG) models. Visual Genome offers a rich variety of concepts and scenes, contributing to the diversity of objects and relationships present in the images. This diversity is crucial for training models that can generalize effectively across a wide range of visual scenarios. Some of the older datasets mentioned in the related works lack the same level of diversity present in Visual Genome. Our goal is to conduct research on continual learning in the context of SGG, and the comprehensive nature of Visual Genome supports this objective. Visual Genome provides a substantial amount of training images, covering numerous objects and relationships. This contrasts with datasets like OpenImages, where the number of objects and relationships is high, but the training samples of individual classes are comparatively limited. Sufficient training samples are essential for effective training, particularly in the context of continual learning where the dataset is divided to study the learning process over multiple tasks. Our emphasis is on continual learning rather than solely addressing the scene graph problem. Thus, we are the first to concentrate specifically on the challenges and dynamics of continual learning in the context of SGG. We have provided novel splits for VG to enable it to facilitate the task of continual learning. Thus, our work builds upon VG to give the first dataset CSEGG, which can be used to study continual learning in the context of SGG.
> > > >
> > > > **LwTF.2 Questions: The caption for Fig 1 suggests objects are nodes and relations are edges. But Fig 1a shows them as nodes of different color. Could you please ensure consistency? Figure 1b intends to show that new objects and relations emerge over time, but some of the uncolored objects (e.g. man, tree) have not appeared previously. So which objects/relations are colored seems arbitrary?**
> > > >
> > > > We appreciate the reviewer for bringing attention to the inconsistency in Fig. 1. In the final manuscript, we will address this issue by ensuring that the color representation in Fig. 1 aligns consistently with the caption, accurately depicting objects as nodes and relations as edges. Additionally, we will revise Fig. 1b to ensure clarity in illustrating the emergence of new objects and relations over time. The updated figure will now accurately reflect the intended representation, addressing the concerns raised by the reviewer.

---

> > > > > ### Author Response · Authors · 2023-11-22
> > > > > **Official Response by Authors**
> > > > >
> > > > > **LwTF.3 Questions: Could you please walk us through the immediate implications of your work for the community, rather than distant possibilities?**
> > > > >
> > > > > * Our paper's primary contribution lies in the design of the Continual Scene Graph Generation (CSEGG) dataset. Unlike a conventional contribution of images or notations, our emphasis is on the thoughtful creation of continual learning scenarios and specific data division within Visual Genome (VG). This tailored dataset enables the analysis of CSEGG models under the challenging conditions posed by the long-tailed nature of the dataset, serving as a valuable resource for further developments and research in continual learning and CSEGG.
> > > > > * Another significant contribution involves evaluating existing continual learning algorithms, such as Replay, Elastic Weight Consolidation (EWC), and PackNet, on established Scene Graph Generation (SGG) architectures. While trends like Replay excelling in mitigating catastrophic forgetting and EWC struggling on longer task sequences were observed in computer vision tasks like Image Classification, we are the first to validate these trends in the context of CSEGG. This work provides insights into the performance of these algorithms in the unique challenges posed by CSEGG.
> > > > > * Additionally, our experiments reveal that continual learning in SGG models enhances model generalizability, as discussed in Sec. 5.3. In Learning Scenario 2 (S2), benchmarking experiments illustrate that while continual learning algorithms perform well in individual incremental object and relationship detection, they face challenges when both are combined. This underscores the need for addressing catastrophic forgetting in scenarios where new objects and relationships are introduced incrementally.
> > > > > * Furthermore, our observations in Sec. 5.2 indicates that existing long-tail sampling techniques are not as effective in CSEGG as they are in SGG. This highlights the necessity of developing long-tailed distribution handling techniques specifically designed for CSEGG.
> > > > >
> > > > > Certainly, our evaluation process not only provided insights into the performance of existing continual learning methods but also inspired the creation of an innovative solution called "Replays via Analysis by Synthesis" (RAS) to tackle the significant challenge of catastrophic forgetting in Continual Scene Graph Generation (CSEGG), explained in detail in Sec. 4.
> > > > >
> > > > > To refine the specifics of our method, ablation studies were conducted, as detailed in Tab. 3. This study aimed to make major design choices for our method, including experiments to select the diversity present in the image along with  identifying the optimal strategy for generating images for the exemplar.
> > > > > * As illustrated in Tab. 3 by comparing (A1 and A2), then (A3 and A4) and finally can also be seen in (A5 and A6), images generated from multiple triplets outperformed those from a single triplet, a trend observed in both Visual Genome (VG) images and the generated images.
> > > > > * Another experiment assessed the quality of notations produced by each task’s model by generating notations for VG images and using them in the exemplar alongside VG images. As shown in Tab. 3 by comparing A2 and A4, applying the same approach to VG images resulted in a minimal drop in Avg.R (16.8%) and F (7%) for the last task of Scenario 1, confirming the quality of the generated notations.
> > > > > * Additional experimentation compared the performance of generated images with VG images, and as demonstrated in Tab. 3 by comparing A4 and A6, the generated images exhibited decent quality. The comparative analysis in Tab. 3 revealed that when using generated images instead of ground truth images, the drop in Avg.R and F was only 7.32% and 6.02%.
> > > > >
> > > > > While a detailed analysis of RAS results can be found in Sec. 5.4 of the main manuscript. This ablation study detailed several key implications such as,
> > > > > * Images with multiple triplets serve better in exemplars than images with single triplet.
> > > > > * Quality of notations produced by each task’s current model is of good quality.
> > > > > * Generated images are of good quality to serve in exemplar. It also shows that the amount of information carried by the generated image is quite similar to the information carried by the VG image.

---

### Official Review · Reviewer_TSA3 · 2023-11-12

**Soundness:** 2 fair
**Presentation:** 1 poor
**Contribution:** 2 fair
**Rating:** 5
**Confidence:** 3

**Summary:**

This paper proposes a scene graph generation dataset for continual learning based on already existing dataset of visual genome. They
proposed three learning scenarios such as incrementing relationships classes, incrementing objects and relationship classes and generalization on relationship between unseen objects over some tasks. They implemented continual learning for scene graph generation over this dataset using a transformer-based approach and a classic two-stage approach and reported their results on 8 evaluation metric including R@K and mR@K.

**Strengths:**

1. Applying continual learning setting on scene graph generation tasks seems to have great potentials for tasks such as robotic navigation etc.
2. Curated a dataset for continual learning setting from an existing benchmark SGG dataset (VG)
3.The codes and dataset will be publicly available

**Weaknesses:**

1. The overall presenation of the paper is difficult to follow and not organized well. (Also too many bold letters for section and figure names)
2. The authors has proposed three learning scenarios and 8 evaluation metrics. All the learning scenarios has multiple tasks (data separations). However, given that there is a lot to report and include, the results are not summarized well in a tabular form. And it is very difficult to follow how each component is contributing in different metrics and leanring scenrios over the tasks.
3. Most of the results are written in textual description. Summarizing them in a tabular form and discussing the interesting finding might help the readers to understand the numbers better.

**Questions:**

The concerns in the weakness section

---

> ### Author Response · Authors · 2023-11-22
> **Official Response by Authors**
>
> **TSA3.1 Weaknesses: The overall presenation of the paper is difficult to follow and not organized well. (Also too many bold letters for section and figure names)**
>
> We apologize to the reviewer for any confusion resulting from the organization and writing of the manuscript. We have addressed the issues in the revised manuscript.
>
> **TSA3.2 Weaknesses: The authors has proposed three learning scenarios and 8 evaluation metrics. All the learning scenarios has multiple tasks (data separations). However, given that there is a lot to report and include, the results are not summarized well in a tabular form. And it is very difficult to follow how each component is contributing in different metrics and leanring scenrios over the tasks.**
>
> We apologize to the reviewer for any confusion resulting from the organization and writing of the manuscript. We have addressed the issues in the revised manuscript. We have included Tab. 1 giving an overview of all learning scenarios.
>
> **TSA3.3 Weaknesses: Most of the results are written in textual description. Summarizing them in a tabular form and discussing the interesting finding might help the readers to understand the numbers better**
>
> We apologize to the reviewer for any confusion resulting from the organization and writing of the manuscript. We have addressed the issues in the revised manuscript. We have also included results of S1 and S2 in tabular form in Tab. 2 with all the metrics.

---

### Author Response · Authors · 2023-11-22
**General comments to all reviewers:**

We thank all the reviewers for the feedback. We provided the response for each individual question below from each reviewer. We also submitted the revised manuscript (both the main text and the supplement). All the figures, tables referenced in the rebuttal are based on the revised manuscript.